

# A systematic look at chromium isotopes in modern shells – implications for paleo-environmental reconstructions

Robert Frei[1][*], Cora Paulukat[1,2], Sylvie Bruggmann[1], Robert M. Klaebe[1]

[1] Department of Geoscience and Natural Resource Management, University of Copenhagen, Øster
Voldgade 10, 1350 Copenhagen K, Denmark

[2] ALS Scandinavia AB, Aurorum 10, 977 75 Luleå, Sweden

[*] Correspondence to: Robert Frei (robertf@ign.ku.dk)

**Abstract.** The chromium isotope system ($^{53}Cr/^{52}Cr$ expressed as $\delta^{53}Cr$ relative to NIST SRM 979) in marine biogenic and non-biogenic carbonates is currently being evaluated as a proxy for the redox state of the ocean. Previous work has concentrated on using corals and foraminifera for this purpose, but investigations focusing on the behavior of Cr in bivalves as potential archives are lacking. Due to their often good preservation, fossil marine biogenic carbonates have the potential to serve as useful archives for the reconstruction of past ocean redox fluctuations and eventually link those to climatic changes throughout Earth's history. Here, we present an evaluation of the Cr isotope system in shells of some modern bivalves. Shell species from Lucidinadae, Cardiidae, Glycimerididae, and Pectenidae, collected systematically from one Mediterranean location (Playa Poniente, Benidorm, Spain) over a three year period, reveal $\delta^{53}Cr$ values ranging from 0.15 to 0.65 ‰, values that are systematically below the local seawater $\delta^{53}Cr$ value of 0.83 +/- 0.05 ‰. This attests for significant reduction of dissolved seawater chromium in the process leading to calcification and thus for control of Cr isotope fractionation during biological routes. A similar, constant offset in $\delta^{53}Cr$ values relative to surface seawater is observed in shells from *Mytilius edulis* from an arctic location (Godhavn, Disko Bay, Greenland). Chromium concentrations in the studied shells are significantly controlled by organic matter and typically range from 0.020 to 0.100 ppm, with some higher concentrations of up to 0.163 ppm recorded in Pectenidae. We also observe subtle, species-dependent differences in average Cr



isotope signatures in the samples from Playa Poniente, particularly of Lucidinadae and Cardiidae, with considerably depressed and elevated $\delta^{53}$Cr values, respectively, relative to the other species investigated. Within-species heterogeneities, both in Cr concentrations and $\delta^{53}$Cr values, are favorably seen to result from vital effects during shell calcification rather than from heterogeneous seawater

composition. This is because we observe that the surface seawater composition in the particular Playa Poniente location remained constant during July month of the three years we collected bivalve samples. Within single shell heterogeneities associated with growth zones reflecting one to several years of growth, both in $\delta^{53}$Cr and Cr concentrations, are observed in a sample of *Placuna placenta* and *Mimachlamys townsendi*. We suspect that these variations are, at least partially, related to

seasonal changes in $\delta^{53}$Cr of surface seawaters. Recognizing the importance of organic substances in the bivalve shells, we propose a model whereby reduction of Cr(VI) originally contained in the seawater as chromate ion and transported to the calcifying space, to Cr(III), is effectively adsorbed onto organic macromolecules which eventually get included in the growing shell carbonates. This study, with its definition of statistically sound offsets in $\delta^{53}$Cr values of certain bivalve species from

ambient seawater, forms a base for futures investigations aimed at using fossil shells as archives for the reconstruction of paleo-seawater redox fluctuations.

## 1. Introduction

Redox processes on land lead to mobilization of Cr from weathering rocks and soils into the run-off. It is now known that oxidation of silicate- and oxide mineral hosted Cr(III), potentially with

catalytic help of $MnO_2$, to Cr(VI) is accompanied by an isotopic fractionation rendering the mobilized Cr(VI) isotopically heavier (Ellis et al., 2002;Zink et al., 2010;Døssing et al., 2011) . Recently, an alternative, redox-independent pathway of Cr mobilization, through ligand-promoted dissolution of Cr-containing solids, was advocated by Saad et al. (2017). This mobilization path bases on the ability of organic acids and siderophores to efficiently bind Cr(III) whereby respective ligand formation is





accompanied by isotope fractionation effects, leading to Cr(III) being enriched in $^{53}$Cr very much like in
redox-dependent mobilization paths.

The fate of Cr transported to the oceans, its transfer/removal to marine sediments and its
cycling through marine organisms, is largely unexplored and complex. Much research focus today is on
the understanding of the redox cycling of Cr in the ocean system, and on investigating marine

sediments and marine organisms as potential archives for recording past redox conditions of the
ocean-atmosphere system through geological time (Frei et al., 2009;Frei et al., 2011;Frei et al.,
2013;Frei et al., 2016;Bonnand et al., 2013;Planavsky et al., 2014;Holmden et al., 2016;D'Arcy et al.,
2017;Rodler et al., 2016a;Rodler et al., 2016b;Gilleaudeau et al., 2016). It is conceivable that the Cr
isotope composition of seawater and marine chemical sediments reflect a complex signal of

oxidation/reduction processes operating within the oceans (Scheiderich et al., 2015;Paulukat et al.,
2016), and it is therefore that one must first understand the individual processes and mechanisms that
governs the transfer of dissolved Cr in seawater into the respective potential archives.

Although naturally depleted in chromium (containing only ppb to ppm levels), marine
carbonates provide a potentially valuable archive of (i) the paleo-seawater $\delta^{53}$Cr composition, and/or

(ii) the past redox conditions of the oceanic water masses. Due to basically continuous geological
record of marine carbonates throughout most of the Earth's history, the Cr isotope studies on marine
carbonate archives are appealing and potentially relevant to addressing the bigger question of the
Earth's oxygenation and temporal evolution of atmospheric $O_2$ levels through geological time.
Nevertheless, as shown by recent studies (Rodler et al., 2015;Pereira et al., 2015;Wang et al., 2016)

the actual mechanism(s) of the redox-controlled isotope fractionation associated with the
incorporation of Cr from seawater into inorganic and biogenic carbonates is rather complex and poorly
understood, thus requiring further systematic investigations in both natural and laboratory-controlled
settings.

Available results from inorganic calcite precipitation experiments revealed that the

incorporation of Cr from a solution into $CaCO_3$ is facilitated as chromate anion ($CrO_4^{2-}$), which replaces




carbonate anion ($CO_3^{2-}$) in the calcite lattice (Tang et al., 2007). This process of inorganic calcification tends to preferentially incorporate heavy $^{53}$Cr isotopes into the mineral, yielding the $\delta^{53}$Cr of calcite that is up to ~0.3 ‰ more positive compared to the fluid; unless the latter is a Cr-poor solution (such as seawater) in which case the isotope fractionation between inorganic calcite and the fluid is

negligible (Rodler et al., 2015).

In contrast, results from biologically produced $CaCO_3$ minerals, such as foraminiferal calcite (Wang et al., 2016) and/or coral aragonite (Pereira et al., 2015), confirmed that these marine organisms produce $CaCO_3$ skeletons that are systematically negatively fractionated , up to ~1 ‰, compared to ambient seawater. Similarly, data by Holmden et al. (2016) from the modern Caribbean

Sea show that the $\delta^{53}$Cr of bulk carbonate sediments is about 0.46 ± 0.14‰ lower relative to local seawater. These results therefore oppose those from inorganic calcite precipitation experiments (cf., Rodler et al. (2015)).

Furthermore, due to a local redox cycling and biological uptake of Cr in the oceans (Semeniuk et al., 2016), the Cr isotope signature of present-day seawater is not globally homogeneous

(Scheiderich et al., 2015; Paulukat et al., 2016). This additionally complicates the application of $\delta^{53}$Cr measurements in marine carbonate archives with respect to deducing information regarding global ocean redox, and implications thereof for climatic changes, on Earth through time. Considering the abovementioned issues and limitations, the full potential of Cr isotopes for paleo-redox studies can only be realized with more detail calibration work done on modern *seawater-carbonate systems* from

different oceanographic settings and locations, where $\delta^{53}$Cr data can be collected simultaneously from (i) local ocean waters, and (ii) precipitated inorganic/biogenic carbonates.

This contribution is a follow-up of a recent study by Farkaš et al. (subm.) who for the first time present a comprehensive Cr isotope investigation of a coupled *seawater-carbonate system* from one of the world's largest carbonate-producing shelf ecosystems, the Great Barrier Reef (Lady Elliot Island,

Australia). These authors present $\delta^{53}$Cr data from local seawaters and selected recent biogenic carbonates (i.e., gastropods, cephalopods, corals, and calcifying algae), complemented by additional



$\delta^{53}$Cr analyses of marine skeletal carbonates (i.e., bivalves, gastropods, and cephalopods) collected from main oceanic water bodies including North and South Atlantic Oceans, North and South Pacific Oceans, and the Mediterranean Sea. A tight coupling between $\delta^{53}$Cr values and cerium anomaly (Ce/Ce*) data from the marine skeletal carbonates led these authors to conclude that marine biogenic carbonates that precipitated from seawater and/or calcifying fluid under the most oxic conditions (i.e., having the most negative Ce/Ce*) can actually reflect the Cr isotope signature of ambient ocean water, shown by a sample of microbial carbonate (i.e., coralline red algae) collected at Lady Elliot Island. Our study goes a step further in that we compare Cr isotope signatures of certain bivalve species from one location in the Mediterranean Sea collected over a period of 3 years with simultaneous collection of surface seawater from that location. This allows us to elaborate on inter-and intra-species Cr isotope variations, with the ultimate aim to eventually deduce systematic fractionation trends/offsets relative to ambient seawater compositions, that could later be used for reconstructing the redox state of past ocean waters.

## 2. Study sites and samples

Bivalve shells (from families Cardiidae, Veneridae, Glycymerididae, Pectinidae, and Lucinidae) and ambient surface seawater samples were collected during the first two July weeks in three successive years from 2015-2017, at Mediterranean Playa Poniente beach, Benidorm, Spain (N38°32'4.20" W0°8'57.30"; Fig.1). In addition, *in situ* growing alive *Mytilus edulis* species and seawater samples were collected by researchers from the Center for Permafrost (CenPerm), University of Copenhagen, at a rocky coast section near arctic Godhavn (Qeqertarsuaq), Disko Island, Greenland (N69°14'44.14" W53°31'38.34"; Fig.1) during fieldwork in June 2016. Respective seawater analyses from the same locations, except the 2017 Playa Poniente sample, were performed earlier and published in Paulukat et al. (2016). Two additional bivalve shells (*Placuna Placenta*; *Mimachlamys townsendi*) from Kakinada Bay, Andhra Pradesh, India (N16°55'30.85" E82°15'43.36"; Fig. 1) and from Hawke's Bay Beach,



Karachi, Pakistan (N24°51'36.46" E66°51'36.66"; Fig. 1), respectively, were used to investigate intra-shell Cr isotope - and Cr concentration ([Cr]) variations. These specimen were cut along a growth

transect into sub-samples and analyzed individually. Pictures of representative shell species (with the sub-sample growth transects of the two specimen studied for intra-shell variations) studied herein are depicted in Figure 2.

Bivalve species studied from Playa Poniente all live in sand in an intertidal setting to about 20m depth. While exact ages of the species studied are not known, based on the relatively small sizes and

number of  annual growth zones in Glycymeris (Beaver et al., 2017;Yamaoka et al., 2016) and *Callista chione* (Moura et al., 2009) we estimate the age range of the majority of shells sampled between ~2-5 years. *Mytilus edulis* lives in the intertidal and sublittoral (up to 5m depth) on a wide range of habitats from rocky shores to estuaries. Our samples were collected from a rocky coast intertidal environment near Godhavn. Growth rates in *Mytilus edulis* are highly variable and dependent on location and

environmental conditions. Typically, under optimal conditions, *Mytilus edulis* can grow up to 60-80mm in length within 2 years (Seed and Suchanek, 1992). *Placuna placenta* (window pane oyster, Capiz) species from Kakinada Bay were purchased from a fisherman who hand-picked them at low tide in a water depth of ~1m. Windowpane oysters from this location have been reported to attain an average length of 122 mm in 1 year and 157 mm in 2 years (Murthy et al., 1979). The *Placuna placenta* sample

studied herein, measuring ~10cm from the apex to the rim (Fig.2), therefore represents about a one year growth period. The Pectenidae species, determined as *Mimachlamys townsendi*, from Hawke's Bay was collected on the sandy beach in 1969 by the lead author himself. There is an extensive variation in growth rates and attained ages of Pectinidae.  Commonly, Mimachlamys has a life time of up to 6 years and in this time reaches sizes between 6-10 cm. Our specimen of *Mimachlamys*

*townsendi* with ~8cm in length therefore represents a fully grown-up shell and our transect (Fig. 2) is representative of several years of growth. These scallops live usually intertidally in shallow water of up to 10 m depth. Some biological and ecological characteristics of scallops can be found in Minchin (2003).



## 3.   Analytical details

### 3.1 Sample preparation and dissolution

Seawater samples were collected into pre-cleaned plastic bottles, filtered through 0.45 $\mu$m nylon membrane filters using a vacuum pump and then acidified and spiked with a $^{50}$Cr-$^{54}$Cr double spike within 1 week from collection.

Shell samples (single shell pieces weighing between 1.5-3 g; up to 7 combined individual shells in case of *Chamelea striatula* and *Loripes lucinalis*) were first physically brushed and washed in Milli-Q™ water (MQ, resistivity 18 M$\Omega$), and then immersed in 2% hydrogen peroxide ($H_2O_2$) for 10 minutes. They were briefly leached in 0.5N HCl and finally thoroughly washed in MQ water. Ten *Mytilus edulis* shells (five dorsal and 5 ventral shells) were combined and powdered in an agate mortar to be used as a so-called "mixed" sample. With the exception of the *Mytilus edulis* samples from Godhavn which were dissolved directly in *aqua regia* after removal of the mussel tissue, pre-cleaned shells (also including 3 samples of *Mytilus edulis* for comparative purposes) were weighed  into chemical porcelain crucibles (CoorsTek™, 15ml capacity) and ashed in a furnace at 750°C for 5 hours prior to dissolution in 6N HCl. The aim with this incineration was to achieve a total dissolution of the respective shells, including the organic material known to have chromium associated with.

### 3.2 Chromium isotope analysis ($\delta^{52}$Cr) by TIMS

Methods used in this study for the purification and isotope analysis of Cr in seawater samples and biogenic carbonates follow those described in Paulukat et al. (2016) and Pereira et al. (2015), with small modifications. Briefly, filtered and spiked seawater samples were transferred into 1 liter Savillex™ teflon beakers, evaporated, re-dissolved in 50ml of *aqua regia*, and evaporated again.



Respective spiking (aiming at a $^{50}Cr/^{52}Cr$ ratio in the sample-spike mixture of between 0.15 to 0.75) of

biogenic carbonates was done during the attack with *aqua regia* or during the 6N HCl attack of

incinerated samples. Spiking prior to ion chromatographic separation procedures enables correction of

any mass-depended Cr isotope fractionation effects that could occur during the chemical purification

and/or mass spectrometric analysis of the samples. The acid-digested and dried down samples (i.e.,

filtered seawaters and pre-cleaned carbonates) were then processed through a two-step Cr

purification chromatography, using a combination of anionic and cationic exchange columns. The first

step used a pass over columns (Spex™) loaded with 2 ml anion exchange resin. The spiked and dried

samples were re-dissolved in ca. 40 ml of 0.1N HCl together with 0.5 ml of a freshly prepared 1N

ammonium persulfate (($(NH_4)_2S_2O_8$; Sigma-Aldrich, BioXtra, ≥98%, lot#MKBR5789V) ) solution, which

acts as an oxidizing agent. The sample solutions, contained in 60 ml Savillex™ teflon vials, were placed

in a microwave oven and heated with closed lids for 50 minutes using a low energy thawing program

to ensure full oxidation of Cr(III) to Cr(IV). After the samples cooled to room temperature, they were

passed through anion exchange columns loaded with 2 ml of pre-cleaned Dowex AG 1 × 8 anion resin

(100-200 mesh). The matrix was washed out with 10 ml of 0.2N HCl, then with 2 ml of 2N HCl and

finally with 5 ml of MQ $H_2O$, before Cr was collected through reduction with 6 ml 2N $HNO_3$ doped with

a few drops of 5% $H_2O_2$. The so-stripped Cr bearing solution was then dried down at 130°C.

The second step used a pass over columns (BioRad™ Econo) loaded with cation exchange resin.

For this, the Cr-bearing samples from the anion columns were re-dissolved in 100 µl of concentrated

HCl and diluted with 2.3 ml ultrapure MilliQ water. This solution was added to the extraction columns

loaded with 2 ml of pre-cleaned Dowex AG50W-X8 cation resin (200-400 mesh). The extraction

procedure principally adhered to that published by Bonnand et al. (2011) and Trinquier et al. (2008)

with only small modifications. The final Cr-bearing liquid cut was dried down at 130°C, ready to be

loaded for Cr isotopic analysis on the thermal ionization mass spectrometer.

Total procedure Cr blanks, including incineration, dissolution, and ion chromatography

procedures remained below 4 ng Cr. In the worst case scenario, using the sample with the lowest [Cr]



in our study (sample Pec-B; [Cr] = 0.021, sample weight = 2.8 g), such blank contribution (assuming the blank Cr composition is of an igneous Earth inventory one), would induce a change in the $\delta^{53}$Cr signature of 0.04 ‰. This is below our current level of analytical precision achieved on the samples studied herein, and below the external reproducibility of between +/- 0.05 to 0.08 ‰ for double-

spiked NIST SRM 979 (see below) under similar measuring conditions. We therefore did not perform a blank correction of our measured sample Cr isotope signatures.

**3.3 Mass spectrometric analyses of Cr**

The Cr isotope measurements were performed on an IsotopX Ltd. PHOENIX thermal ionization mass spectrometer (TIMS) equipped with eight Faraday collectors that allow simultaneous collection of the four chromium beams (50Cr+, 52Cr+, 53Cr+, 54Cr+) together with interfering 49Ti+, 51V+, and 56Fe+ masses.

The separated Cr residues were loaded onto outgassed Re-filaments using a loading solution

consisting of 1 µl of 0.5N $H_3PO_4$, 2.5 µl silicic acid (Gerstenberger and Haase, 1997), and 0.5 µl of 0.5N $H_3BO_3$. The samples were analyzed at temperatures between 1050-1250°C and 52Cr beam intensities of between 0.35 and 1V. One run consisted of 120 cycles and where possible, every sample was run at least twice. The final $\delta^{53}$Cr values of the samples were determined as the average of the repeated analysis and are reported in per mil (‰) with ± standard deviation (2σ) relative to the international

standard reference material NIST SRM 979 as:

$$\delta^{53}\text{Cr (‰)} = [(^{53}\text{Cr}/^{52}\text{Cr})_{SAMPLE} / (^{53}\text{Cr}/^{52}\text{Cr})_{NIST\ SRM979} - 1] \times 1000$$

The within-run two standard errors of the measurements reported in this study were

consistently ≤ 0.1‰. The external reproducibility was determined using average $\delta^{53}$Cr values of double spiked NIST SRM 979 measured under the same conditions as the samples on the Phoenix. Figure 3



depicts the averages of 10 runs each from the same filament loaded with 200 ng of double spiked NIST SRM 979 at beam intensities of 0.35V and of 1V. The external reproducibility of the standard under these conditions was +/- 0.08‰ and 0.05‰ ($2\sigma$), respectively, at the above mentioned $^{53}$Cr beam

intensities (Fig. 3). The average composition of the 0.35V and 1V multiple NIST SRM 979 runs analyzed during the course of this study shows an average offset of +0.04 ± 0.03‰ ($2\sigma$; n=11; Fig. 3) on our machine compared to the 0‰ certified value of this standards. This offset stems from the original calibration of our double spike relative to the NIST 3112a Cr standard, and the observed offset of 0.04‰ was deducted from the raw $\delta^{53}$Cr results to account for this small discrepancy.


## 4. Results

### 4.1. Surface seawater - Chromium isotope compositions ($\delta^{53}$Cr) and elemental concentrations

Table 1 lists the Cr isotope compositions and Cr concentrations the of surface seawater
samples relevant to this study. Waters collected during 4 subsequent years in July from Playa Poniente yield surprisingly homogeneous Cr isotope compositions and dissolved [Cr] that range from $\delta^{53}$Cr = 0.81 – 0.85‰, and from 222 – 280 ng/kg, respectively. These are comparable with surface seawater data ($\delta^{53}$Cr = 0.81 – 0.96‰, [Cr] = 239-306 ng/kg) collected from Playa Albir, a beach situated ca. 9.5 km to the ENE of Playa Poniente (Fig. 1), in the years 2013 through 2015 (data published by Paulukat
et al. (2016)). The Playa Poniente data are compatible with data from other Mediterranean surface seawaters, which distinguish in [Cr] vs. $\delta^{53}$Cr space from Baltic Sea seawater, but are compatible with the trend of an inverse logarithmic relationship between $\delta^{53}$Cr and [Cr] defined by Scheiderich et al. (2015) and substantiated later by Paulukat et al. (2016) of worldwide Atlantic and Pacific ocean waters. Three separate analyses of seawater from Disko Island, published by Paulukat et al. (2016), are
characterized by slightly lower [Cr] and $\delta^{53}$Cr values compared to the Mediterranean waters, but are similar to other waters from the North Atlantic (c.f., Fig. 2 in Paulukat et al. (2016)).





## 4.2 Shells - Chromium isotope compositions ($\delta^{53}$Cr) and chromium concentrations

260       Cr isotope compositions and [Cr] of a variety of bivalve species from Playa Poniente (an undefined species of Cardiidae, *Callista chione*, *Chamelea gallina*, *Chamelea striulata*, an undefined species of Glycymeris, *Loripes lucinalis*, *Pecten jacobaeus*, *Venus verrucosa*, *Venus nux*, and *Arca Navicularis*), of *Mytilus edulis* species from Godhavn, and of sub-samples from a window pane oyster (*Placuna placenta*) from Kakinada Bay and of a specimen of *Mimachlamys townsendi* (Pectenidae)

from Hawke's Bay Beach are listed in Table 2. Respective data are plotted in Figs. 4-7, together with the ranges of local surface seawaters and the range of igneous Earth reservoirs as defined by Schoenberg et al. (2008). There is no obvious correlation between $\delta^{53}$Cr and [Cr] data ($r^2$= 0.18), diagram not shown) of the samples analyzed herein.

      Bivalves collected from Playa Poniente generally show low and scattered Cr concentrations

ranging from 0.027 to 0.163 ppm, with the largest variations recorded in Glycymeris, and the systematically highest concentrations measured in species of *Pecten jacobaeus* (Fig. 4; Table 2). Isotopically, the assembly of shell data from Playa Poniente point to a rather restricted compositional band with $\delta^{53}$Cr data ranging from 0.157 to 0.636 ‰, significantly lower than the local surface seawater average over four consecutive years of $\delta^{53}$Cr = 0.83 +/- 0.05 ‰ (Fig. 4). On closer inspection,

we however note some distinctive Cr isotope ranges defined by the different bivalve species analyzed. So, for example, samples from Cardiidae exhibit the highest ($\delta^{53}$Cr = 0.60+/-0.13 ‰) -, while samples from *Loripes lucinalis* show the lowest Cr isotope compositions ($\delta^{53}$Cr = 0.21 +/- 0.10 ‰; Table 2, Fig. 4). One subsample of *Loripes lucinalis* (sample PP 15 D unashed; Table 2) was dissolved in *aqua regia* without prior incineration. This specific sample yielded a distinctively lower Cr concentration ([Cr] =

0.027 ppm) compared to all other *Loripes lucinalis* samples, but at the same time a $\delta^{53}$Cr value which statistically cannot be distinguished from the other *Loripes lucinalis* samples (Fig. 4). While this, as expected, points to the fact that a significant fraction (in fact roughly 50% in case of *Loripes lucinalis)* of the total Cr budget in biogenic carbonates is associated with organic material, and not with



carbonate itself, it also points to the likelihood that there is not much difference in the Cr isotope

composition of these two potential Cr host materials. This result is substantiated and supported by our

study of *Mytilus edulis* from artic Godhavn (see below). Last not but least, we do not see any

statistically significant and systematic differences in Cr isotope compositions and Cr concentrations

between bivalve species collected during the consecutive sampling years. This conforms to the rather

homogeneous surface seawater compositions analyzed from Playa Poniente and the neighboring

location Playa Albir during the entire sampling period (Table 1).

Results of entirely *aqua regia* dissolved half-shells of *Mytilus edulis* from Godhavn in the Disko

Bay, Greenland, are plotted in a similar combined $\delta^{53}$Cr – [Cr] diagram as the shells from

Mediterranean Playa Poniente in Figure 5. These analyses are complemented by a bulk analysis of

powdered multiple *Mytilus edulis* specimen from the same location. In addition, two specimen (in

each case the dorsal or ventral shell counterparts of the respective *Mytilus edulis* specimen dissolved

by *aqua regia*; i.e. samples God-4 and God-5; Table 2) were ashed before final dissolution, in order to

evaluate the importance of Cr associated with organic material in these shells compared to the total Cr

budget. This was also done with an aliquot of the powdered *Mytilus edulis* mix. All data define an

average $\delta^{53}$Cr value of 0.11 +/- 0.05 ‰ (n=10; 2$\sigma$), significantly lower than the local surface seawater

of 0.73 +/- 0.05 ‰ (Paulukat et al. (2016); Fig. 5). [Cr] for *aqua regia* dissolved specimen are a bit more

variable, defining an average of [Cr] = 0.039 +/- 0.009 ppm (n=7, 2$\sigma$). Again, the incinerated aliquots,

while isotopically not distinguishable from the unashed samples, yielded about twice as high a [Cr]

([Cr] = 0.068 +/- 0.004 ppm (n=3, 2$\sigma$)). As mentioned above, this indicates that organic material,

effectively attacked by incineration, is a major and significant host of Cr in these bivalves, besides the

biogenic carbonate. Again, as already emphasized by the results of *Loripes lucinalis* from Playa

Poniente, the two Cr host materials seem not to be isotopically distinguishable. While, with the

exception of *Loripes lucinalis*, the negative offset ($\Delta_{Cr}$) of different bivalve species from local surface

seawater at Playa Poniente is between ~0.4 to ~0.2 ‰, the respective $\Delta_{Cr}$ value for *Mytilus edulis* from



Godhavn is higher, ~0.7‰, and comparable to $\Delta_{Cr}$ ~0.6‰ defined by *Loripes lucinalis* from Playa

Poniente (see details below).

Results from two profiles along respective major growth transects of a specimen of *Placuna*

*placenta* from Kakinada Bay, Andhra Pradesh, India, and a species of Pectinidae (*Mimachlamys*

*townsendi*) from Hawke's Bay, Karachi, Pakistan, are plotted in Figure 6 (*Placuna placenta*) and Figure

7 (*Mimachlamys townsendi*).

[Cr] along the *Placuna placenta* growth profile vary from 0.03 to 0.25 ppm, but this variation is

much smaller if the first sample (Cap-A) is excluded. Sample Cap-A incorporates the beak of the shell

and a first growth zone which is visually characterized by a more brown (organic-rich) color (Fig. 3). We

here attribute the exceptionally high Cr concentration in this sample to increased, organic-rich

components which seem to act as efficient Cr hosts, basically reflecting our experiments on *Mytilus*

*edulis* from Godhavn (Fig. 5). Although some fluctuations in [Cr] and d53Cr values beyond the

statistical errors across the beak-margin profile of the studied *Placuna placenta* specimen exist (which

we may attribute to local (seasonal?) changes in seawater composition during growth and/or to

changing reductive efficiencies during the calcification process), the studied specimen pretty much

averages such environmental and biogenic changes out over its entire growth period estimated to be

about 2 years. Chromium concentrations in this oyster from the Indian Ocean are comparable with

those of the Mediterranean shells studied from Playa Poniente (Table 2, Fig. 4). In contrast, $\delta^{53}$Cr

values of *Placuna placenta* are lower than those recorded in the Mediterranean bivalves and show

values that are just about statistically distinguishable from the igneous Earth inventory value of -0.12

+/- 0.11 ‰ defined by Schoenberg et al. (2008). Lack of a respective surface seawater sample from

Kakinada Bay itself does not allow for a concrete definition of the $\Delta_{Cr}$ value- the nearest surface

seawater sample from which we have a Cr isotope composition available is from the Bay of Bengal

with $\delta^{53}$Cr = 0.55 +/- 0.08 ‰ (Paulukat et al., 2015). If we assume that the local surface seawater in

Kakinada Bay has a similar Cr isotope composition, then $\Delta_{Cr}$ would be ~0.5‰, an offset which is similar



to that of *Loripes lucinalis* from Playa Poniente, but higher than most other species from this Mediterranean location.

        Intra-species variations of $\delta^{53}$Cr and [Cr] of *Mimachlamys townsendi* from Hawke's Bay are depicted in Figure 7. The average $\delta^{53}$Cr value of all 8 profile samples is 0.07+/-0.11 ‰ (2$\sigma$) and statistically indistinguishable from that of *Placuna placenta* (average $\delta^{53}$Cr = 0.05 +/- 0.19 ‰, n= 7, 2$\sigma$).

With the exception, like in *Placuna placenta*, of the sample closest to the apex/hinge of the shell (sample Pec-H; Fig. 7)) which yielded by far the highest [Cr] in the profile, the [Cr] of the remaining profile samples are ~0.04 ppm. In the specimen studied, the d53Cr values along the profile are statistically indistinguishable from each other. If Hawke's Bay surface seawater has a $\delta^{53}$Cr similar to that of the Bengal Bay (Paulukat et al., (2015); and assuming it has remained about the same since the

collection of the *Mimachlamys townsendi* sample), then *Mimachlamys townsendi* exhibits the same $\Delta_{cr}$ value (~0.5 ‰) as *Placuna placenta* from Kakinada Bay. If calcification processes in *Mimachlamys townsendi* remained constant in terms of biogenic reduction of Cr(VI) to isotopically lighter Cr(III) over the several years of growth of the specimen studied, then this would signify are more or less constant d53Cr of surface water in this location.


## 5. Discussion

### 5.1 Present state of knowledge of the behavior of chromium in the marine biogenic carbonate system

Chromium-isotope compositions of recent and ancient skeletal and non-skeletal carbonates are currently explored as a (paleo-) redox-proxy for shallow seawater (corals: Pereira et al.(2015); foraminifera: Wang et al.(2016); calcifying algae, molluscs, corals: Farkaš et al. (subm.)). The idea behind this approach is that biogenic and non-biogenic carbonates could potentially be used as archives recording the Cr-isotope composition of seawater in which they formed, and with this

contribute to the reconstruction of past paleo-environmental changes in the marine realm that may





have potentially resulted from climate changes on land. However, investigations addressing the

behavior and uptake mechanism of Cr, and the potential isotope fractionations between seawater and

biogenic carbonates are scarce. All studies so far conducted on marine biogenic carbonates have

revealed the incorporation of isotopically lighter chromium into skeletal and non-skeletal carbonates

compared to Cr isotope signatures of seawater at the respective sampling sites (e.g., Pereira et al.

(2015), Holmden et al. (2016); Farkaš et al., (subm.)). Due to lack of ambient seawater $\delta^{53}$Cr data to

compare the $^{53}$Cr data of various foraminifera analyzed by Wang et al. (2016), conclusion with respect

to using $\delta^{53}$Cr data of foraminiferal species as reliable proxy of seawater $\delta^{53}$Cr could not be made in

the respective study. However, these authors observed large $\delta^{53}$Cr variations between species within

and among samples. Such variations in $\delta^{53}$Cr among different samples could be explained by

heterogeneous seawater $\delta^{53}$Cr However, Wang et al. (2016) also found that foraminifera species with

similar depth habitats from the same core- top sample also yielded different $\delta^{53}$Cr values. In addition,

within samples, foraminifera with shallower habitats yielded consistently lower $\delta^{53}$Cr than those with

deeper habitats, which these authors correctly described as opposite to the general patterns expected

in seawater $\delta^{53}$Cr (Bonnand et al. (2013); Scheiderich et al.(2015); Paulukat et al. (2016)). The study of

Farkaš et al.(subm.) deals with chromium isotope variations in recent biogenic carbonates and ocean

waters from Lady Elliot Island located in the southern Great Barrier Reef, Australia. The Cr isotope data

from the Lady Elliot Island seawater-carbonate system, representing the South Pacific region, were

complemented by $\delta^{53}$Cr analyses of recent skeletal carbonates originating from the North Pacific,

North and South Atlantic Oceans, and from the Mediterranean Sea. The results of Farkaš et al. (subm.),

combined with the published seawater $\delta^{53}$Cr data from the above oceanic water bodies, confirm the

results of (Pereira et al., 2015) that marine biogenic carbonates are systematically enriched in light Cr

isotopes compared to ambient ocean waters. There is growing debate about the mechanisms inherent

to Cr isotope fractionation during calcifying processes:  Results published so far point into a direction

whereby vital processes (i.e., biology) could potentially play a major role in controlling Cr isotope

fractionation during skeletal, foraminiferal, algal and shell calcification. The apparent variability of





foraminiferal $\delta^{53}$Cr values in the study of Wang et al. (2016) could be envisaged as to result from

variable Cr uptake mechanisms. These authors propose, that in regions with high dissolved organic

matter, foraminifera could preferentially uptake Cr(III) associated with dissolved organic phases

and/or organic matter, as observed for some phytoplankton (Semeniuk et al., 2016). In regions where

dissolved organic concentration is low, foraminifera may switch to the reductive Cr(VI) uptake

mechanism, as proposed for coral growth (Pereira et al., 2015). As Cr(III) is typically isotopically lighter

than Cr(VI) in both equilibrium and kinetic fractionations (e.g., Ellis et al. (2002); Schauble et al. (2004);

Wang et al. (2015)), Cr(III) uptake mechanism via organic matter would lead to relatively low $\delta^{53}$Cr

values. A reductive Cr(VI) uptake mechanism is also expected to lead to lower- than- seawater $\delta^{53}$Cr

values in marine biogenic carbonate systems. In this case, the exact $\delta^{53}$Cr value would depend on the

extent of reduction and specific metabolism. A small extent of reduction would lead to low $\delta^{53}$Cr

values, while quantitative reduction would lead to similar to seawater values (Wang et al., 2016).

Direct incorporation of organic acid and/or siderophore bound Cr(III), as recently proposed by Saad et

al. (2017) to have a significant impact on the Cr cycle via their release from the continents to the

oceans, can also be considered to play a role in the marine biogenic calcification processes as these

compounds have been shown to carry isotopically heavy Cr(III) compositions that are reached through

redox-independent chromium isotope fractionation induced by ligand-promoted Cr(III) dissolution on

land.


**5.2. Chromium distribution coefficients (D$_{Cr}$) between bivalve shell carbonates and seawater**

Our sample sets from Playa Poniente and from Godhavn, which contain both surface seawater

and bivalve shell data, allow for a direct calculation of the distribution coefficients (D$_{Cr}$) describing the

partitioning of chromium between biogenic CaCO$_3$ and seawater at the respective study sites. The D$_{Cr}$

is calculated as:



$$D_{Cr} = ([Cr]_{CaCO3} / [Cr]_{seawater})$$


where $[Cr]_{CaCO3}$ represents the measured total concentration of chromium in the bivalve shell (CaCO3 and organic matter hosted) and $[Cr]_{seawater}$ the measured dissolved chromium concentration of the surface seawater at the respective location (e.g., 0.000300 ppm; 0.000176 ppm, and 0.000254 ppm, respectively, for Hawke's Bay/Kakinada Bay, Godhavn and Playa Poniente).

The calculated $D_{Cr}$ values for biogenic carbonates are listed in Table 2.  Our data span a wide range with values from 70 to 1297, but the upper data limit is characterized by a few exceptionally high $D_{Cr}$ values, f.e. that of sample Cap A ($D_{Cr}$ = 1297) and Pec H ($D_{Cr}$ = 820) from the respective hinges of the *Placuna placenta* species from Kakinada Bay and the *Mimachlamys townsendi* specimen from Hawke's Bay which probably are characterized by elevated organic matter. By far most samples have a

more restricted $D_{Cr}$ range with values between 70 and 640. The $D_{Cr}$ range presented herein for bivalves is much more narrow compared to data from the study of Farkaš et al. (subm.) in which these authors present $D_{Cr}$ values spanning more than three orders of magnitude (from 79 up to 10895) in marine biogenic carbonates from Lady Elliot Island and other worldwide locations. However, as Farkaš et al.(subm.) note, skeletal carbonates (i.e., corals, molluscs) in their study tend to have systematically

lower values (from ~80 to ~780) than microbial carbonates (i.e., calcifying algae) that yielded much higher $D_{Cr}$ of ~1000 and 2356. The range of $D_{Cr}$ values for corals and molluscs in the study of Farkaš et al. (subm.) otherwise compares well with the range of $D_{Cr}$ values for bivalve shells in our study, and are within the range of $D_{Cr}$ calculated for formanifera that vary from ~300 to 4000 (Wang et al., 2016) and with $D_{Cr}$ values for corals in the range of 135 to 253 calculated from data in Pereira et al. (2015). Such

high $D_{Cr}$ values observed in biogenic carbonates produced by different marine organisms point to a strong biological control over the incorporation of Cr from seawater into $CaCO_3$ skeletons, where it could be incorporated either as Cr(III) and/or Cr(VI) depending on species-specific redox cycling of Cr





(cf., Wang et al. (2016); Semeniuk et al. (2016)) and/or, as recently suggested, directly assimilated as organic ligand-bound Cr during biological uptake (Saad et al., 2017).


**5.3 Evidence for isotopically fractioned Cr incorporated in bivalve shell carbonates**

Our study confirms the outcome of previous investigations (Wang et al., 2016;Pereira et al., 2015;Farkaš et al., subm.) showing that marine (skeletal and non-skeletal) biogenic carbonates are

characterized by isotopically variably fractionated, but systematically [53]Cr enriched Cr compositions that have $\delta^{53}$Cr values above the Earth's igneous inventory value of 0.12 +/- 0.11 ‰. From data which allow direct comparison with ambient seawater compositions, including those presented in this study, it can also be deduced that the biogenic carbonates so far analyzed all have Cr isotope compositions which are depleted in [53]Cr relative to respective seawater values, implying redox cycling, in particular

reductive processes, to take place somewhere during the uptake and calcification processes.

In order to explain the isotopically light Cr incorporated in coral skeletal carbonate, Pereira et al. (2015) propose a mechanism whereby initial photoreduction of isotopically heavy Cr(VI) in the surface seawater to isotopically lighter Cr(III) in the endodermal layer of corals must be followed by efficient and effective re-oxidation of reduced Cr species to favor subsequent chromate ($CrO_4^{2-}$)

substitution during the calcifying processes ultimately leading to the coral skeleton.

To further test a possible role of biologically mediated redox processes during the incorporation of Cr from seawater into biogenic carbonate, Farkaš et al.(subm.) investigated the relationship between $\delta^{53}$Cr and cerium anomaly (Ce/Ce*) data in their biogenic carbonate samples from Lady Eliot Island. The strong and statistically significant negative correlation between Ce/Ce* and

$\delta^{53}$Cr data in their study, and the fact that the intercept of the $\delta^{53}$Cr vs. Ce/Ce* correlation line overlaps with the Cr isotope composition of a local ocean water, led these authors to suggest that marine biogenic carbonates that precipitated from seawater and/or calcifying fluid under the most



oxic conditions (i.e., having the most negative Ce/Ce*) can actually reflect the Cr isotope signature of ambient ocean water.


## 5.4 Biomineralization / calcification and incorporation of chromium into shell

A central question is that regarding the mechanisms on how dissolved chromium from the seawater behaves during biomineralization and calcification processes, and ultimately is

incorporated into marine biogenic carbonates. It becomes evident from recent studies (Wang et al., 2016;Pereira et al., 2015;Farkaš et al., subm.) that redox mediated processes play a role during calcification because marine biogenic carbonates measured so far all are characterized by significantly 53Cr depleted (i.e., isotopically lighter) signatures relative to ambient seawaters. Reduction processes of dissolved Cr(VI) complexes to Cr(III) species in ocean water have been used by Scheiderich et al.

(2015) and Paulukat et al. (2016) to explain the $\delta^{53}$Cr variations in world's oceans. Scavenging of isotopically light Cr(III) to deeper water and sediment, potentially by phytoplankton (Semeniuk et al., 2016), and subsequent release of this seawater-derived Cr(III) back into seawater, either as organic complexes with Cr(III) or after oxidation to Cr(VI), are advocated as potential processes to explain the $\delta^{53}$Cr *vs* [Cr] fractionation trend in seawater.

It is unclear, whether Cr can be directly incorporated into the carbonate structures as Cr(VI) forming part of $CrO_4^{2-}$ compounds, or whether reduced species of Cr(III) can be assimilated/adsorbed or structurally bound into skeletal carbonates or associate with a multitude of known organic matrices contained within and along cleave/grain boundaries of calcifying layer carbonates. In one way or the other, models that address the mechanisms of Cr uptake during

calcification processes need to involve the fact that bulk marine biogenic carbonates are isotopically lighter than ambient seawater in which they are formed. Pereira et al. (2015) proposed a model for skeletal carbonates of corals whereby initial photoreduction of isotopically heavy Cr(VI) to isotopically lighter Cr(III) in the endodermal layer of corals must be followed by efficient and effective re-oxidation



of reduced Cr species to favor subsequent chromate ($CrO_4^{2-}$) substitution during the calcifying

processes ultimately leading to the formation of the coral skeleton.

A vast number of studies dedicated to biomineralization processes of marine biogenic carbonate producers have recognized the importance of organic network matrices (Griesshaber et al., 2013), and of organic macromolecules in particular (e.g., (Suzuki et al., 2011;Okumura et al., 2013), in the organic–inorganic interaction in biomineralization of, particularly, molluscan shells. Major

components of the shell are calcium carbonate, which ordinarily exists as a crystalline polymorph, either calcite or aragonite. The type of polymorph, crystal orientation, morphology and texture of the crystals are regulated in the shell. Studies have shown that shells are not composed of purely inorganic carbonate crystals, but contain small amounts of organic substances to regulate the structure and property of these crystals (e.g., (Falini et al., 1996;Belcher et al., 1996;Okumura et al., 2013). Suzuki et

al. (2011) visualized intra-crystalline spherular structures in shell carbonates containing carbon from organic macromolecules. The size of the spherules identified by these authors roughly corresponded to that of soluble organic macromolecules that these authors extracted from the nacreous layer (innermost layer of the shell of a mollusk secreted by the mantle epithelium layer). Their function for the crystal formation of molluscan shells remains unclear though. A comprehensive review on the

presence and role of organic matrices for the growth of mollusk shells is contained in Suzuki and Nagasawa (2013).

While recognizing that individual processes leading to the presence of various organic compounds in mollusk shell calcification and growth, and biomineralization processes are complex and topic of intensified research in the last years, we like to focus our attention on the potential role of

such organic matrices as hosts for Cr in mollusk shells. A hint that organics may play a defining role stems from our few results which compare [Cr] in shell material that has been incinerated to corresponding [Cr] in aliquots which were attacked with *aqua regia* to preferentially attack the carbonate. While it is clear from our study of *Mytilus edulis* from Godhavn with an apparent organic-rich periostracum that this outermost shell layer itself may contain elevated [Cr], based on a similar



result (c.f., Fig. 3, Table 2) performed on *Loripes lucinalis* from Playa Poniente where the actual

periostractum has been mechanically removed by tidal abrasion in the beach sand, we suspect that

organics contained in the nacreous layer are equally important as potential Cr hosts. In all cases (see

above and Table 2) we note significantly higher [Cr] in the ashed samples, which we see as a

consequence of effective release of organic-material bound Cr (otherwise only weakly- or even not

attacked at all by the hydrochloric acid) during burning of the organic material. So, for example, Suzuki

et al. (2007) hydrolyzed the insoluble organic matrices from the prismatic layer in the mollusk shell

with 6 M HCl. These authors detected D-glucosamine hydrochloride, known as a degradation product

of chitin, using nuclear magnetic resonance spectroscopy measurements. In the nacreous layer of

mollusk shells, chitin serves as the major component of the organic framework, building up the

compartment structure and controlling the morphology of calcium carbonate crystals (Falini and

Fermani, 2004).

## 5.5  A model explaining the occurrence of Cr in molluscan shells

Adopting the schematic framework that includes the representation of the localization and

function of organic matrices with respect to calcium carbonate crystals in the nacreous layer of

mollusks we like to propose a model that explains the transfer of Cr from the water into the calcifying

space and the incorporation of Cr into shell carbonates (Fig. 8). During adult shell formation, the

periostracum, which is not mineralized and covers the external surface of the shell, is formed first, and

the calcified layer subsequently forms on the periostracum (e.g., (Checa, 2000)). The shell is in contact

with the mantle, which supplies the periostracum and calcified layers with inorganic ions and organic

matrices through the extrapallial fluid (for a review, see Marin et al. (2012)). This extrapallial space is

supposed to be the confined medium where all the ingredients for calcification self-assemble. The

extrapallial fluid, filling this space, is supersaturated with respect to calcium carbonate. In addition to

the precursors ions for mineralization - calcium and bicarbonate - this fluid contains several other

inorganic ions, such as $Na^+$, $K^+$, $Mg^{2+}$, $Cl^-$ and $SO_4^{2-}$, and minor elements, such as Sr and Fe. Its pH is usually slightly basic, in the range of 7.4 – 8.3, for marine and freshwater mollusks (Marin et al., 2012). This fluid also contains organic molecules. As the fluid is supersaturated, these macromolecules – in particular acidic proteins and GAGs (group specific antigen) - are supposed to transiently maintain

calcium in solution, by inhibiting the precipitation of calcium carbonate, and by allowing it to precipitate where needed (Marin et al., 2012). The manner the inorganic precursors of calcification are driven to the site of mineralization is still speculative. Figure 8 schematically shows the growth front in an interlamellar space of the nacreous layer, confined by chitinous membranes, as proposed by Suzuki and Nagasawa (2013). We emphasize that under neutral to basic pH, as inferred for an extrapallial

fluid, Cr is present either as dissolved Cr(VI) compound, as Cr(III) species adsorbed onto organic macromolecules and/or as dissolved organic substances. The fact that $\delta^{53}$Cr measured in bivalve shells is systematically lower than ambient seawater implies that reduction of dissolved Cr(VI) in seawater, transferred to the calcifying space , is likely promoted by the organic macromolecules, which are densely localized on the surface of the interlamellar membranes (Suzuki and Nagasawa, 2013;Suzuki et

al., 2011). So-formed isotopically light Cr(III) species, effectively adsorbed onto organic macromolecules, adhere to the chitinous membranes where they are incorporated inside growing carbonate crystals filling the space, whereas other organic molecules cover the surface of these crystals. Finally, surface-covering organic macromolecules are also incorporated inside the crystal when the space is filled with the crystal. Some Cr might also be directly incorporated into the

carbonate lattices during growth, where chromate ions may coprecipitate with calcite (Tang et al., 2007). In such a scenario, the measured bulk $\delta^{53}$Cr values of mollusk shells would reflect a mixture of both Cr(VI) and Cr(III) characteristic of the ambient seawater and an isotopically lighter, Cr(III) fraction ultimately associated with the organic molecules in the shells. The exact $\delta^{53}$Cr value would depend on the extent of reduction and specific metabolism. A small extent of reduction would lead to low $\delta^{53}$Cr

values while quantitative reduction of dissolved Cr(VI) would lead to similar to seawater values.





## 5.6 Inter-and intraspecies shell variations

While from the studies conducted earlier (e.g., (Pereira et al., 2015;Wang et al., 2016)

and from this study it is now evident that marine biogenic carbonates are characterized by $\delta^{53}$Cr

values that are less fractionated in comparison to ambient seawater, it remains unclear whether these

isotopic offsets are species dependent. Wang et al. (2016) observed large $\delta^{53}$Cr variations between

foraminifera species within and among samples. As advocated by these authors, the variation in $\delta^{53}$Cr

among different samples of the same species could be explained by heterogeneous seawater $\delta^{53}$Cr.

However, Wang et al. (2016) also found that foraminifera species with similar depth habitats from the

same core- top sample also yielded different $\delta^{53}$Cr values. Species dependent $\delta^{53}$Cr variations are

furthermore complicated by the observation that species with shallower water depth habitats yielded

consistently lower $\delta^{53}$Cr than species preferring deeper water environments, which is opposite to the

general patterns expected in seawater $\delta^{53}$Cr (Bonnand et al., 2013;Scheiderich et al., 2015). These

observations hint at the possibility that species-dependent biological (metabolic) processes may play a

major control on Cr isotope fractionation during biomineralization/calcification processes of marine

biogenic carbonate producers in general, not only in foraminiferal calcification. Our data herein

contribute to a more systematic assessment of the above: The systematic sampling of some bivalve

species from the same location over several years, together with respective ambient surface

seawaters, reveals that subtle inter-species differences of average bulk $\delta^{53}$Cr signatures exist amongst

different species. While 5 species (*Calista chione*, an unidentified species of Cardiidae, *Chamelea*

*striulata*, *Glycymeris glycymeris*, and *Pecten jacobaeus*) however, at the 2$\sigma$ level, cannot be

statistically distinguished by their average $\delta^{53}$Cr values (Fig. 3), *Loripes luncinalis* is an exception and

yielded, at average, lower $\delta^{53}$Cr values than the other species. Thus, while we observe subtle

differences in the average $\delta^{53}$Cr signatures of individual bivalve species from Playa Poniente, intra-

species variations, as observed by Wang et al. (2016) for certain foraminifera, are statistically not

discernable. The exception to this are two samples of *Arca Navicularis*, sampled simultaneously in





2015, which both show distinctly different $\delta^{53}$Cr signatures of 0.570 and 0.166 ‰, and also significantly different [Cr] of 0.052 and 0.166 ppm, respectively. We are unable, at this point, to

explain these discrepancies observed in *Arca Navicularis*. Last not but least, while [Cr] in the samples studied scatter considerably between ~0.03 to 0.10 ppm and do not correlate with bivalve species, there is an exception to this which is reflected by the data of *Pecten jacobaeus*. The three samples of this species all revealed elevated [Cr] in the range of 0.127 to 0.163 ppm (Table 2, Fig. 3). Whether or not the intensity of pigmentation (*Pecten jacobaeus* shows a red pigmentation that increases from the

hinge to the margin of the shell; Fig. 2) is not clear, but it could partially explain the increased [Cr] scatter in the analyses from *Glycymeris glycymeris* (c.f., Fig. 2, Table 2) which exhibits similar variations in pigmentation amongst individual samples. Importantly, however, is the fact that the increased scatter of [Cr] does not seem to translate into an increased scatter of bulk $\delta^{53}$Cr values of the bivalve shells studies, nor does [Cr] seem to correlate with $\delta^{53}$Cr in any of the species studied either. If, as

emphasized in our preferred scenario, that [Cr] in the bivalve shell is significantly associated with organic matter, it implies that intralamellar reductive processes eventually leading to adsorption of isotopically light Cr(III) onto organic macromolecules, and the production rate of these macromolecules, are likely metabolically controlled/buffered prior to their encapsulation into the shell carbonates. This is maybe best exemplified by the *Mytilus edulis* sample suite from Godhavn. This suite

of samples reveals limited intra-species variations both in $\delta^{53}$Cr and [Cr] among the 6 half-shells analyzed, which we take as an indication for an effective and stabilizing biological control, potentially via organic macromolecule production, of biomineralization processes in general, and of Cr incorporation into the shell carbonates.

Our study may eventually also contribute to the understanding of the environmental

stability over relevant growth periods (several years) around the calcifying space of bivalves. However, such investigations are dependent on the knowledge of the seawater Cr isotope composition during the respective growth periods (in our case during growth of the *Placuna placenta* from Kakinada Bay and the *Mimachlamys townsendi* sample form Hawke's Beach, which we do not have at hand. It is




strongly perceivable that surface seawater conditions at a specific location are not, and have not been

constant, and this has been shown for the $\delta^{53}$Cr values of surface water from the Baltic Sea by

Paulukat et al. (2016). These authors correlated seasonal fluctuations in $\delta^{53}$Cr with algae bloom

periods, and thus with the seasonal presence of strong Cr(VI) reducers capable of depleting the [Cr] in

the surface waters considerably by reductive adsorption mechanisms. Seasonal fluctuations could

explain the sinusoidal $\delta^{53}$Cr growth pattern in the studied *Placuna placenta* shell (Fig. 6) whose size

roughly implies a ~1 year's growth period. Likewise, small fluctuations in *Mimachlamys townsendi* of

$\delta^{53}$Cr signatures over the entire growth period of the specimen studied could reflect seasonal changes

of the ambient surface seawater during this several years long growth period. We however want to

emphasize that these within- shell $\delta^{53}$Cr fluctuations, in the order of +/- 0.15‰, compare well with

inter-species fluctuations of the same order observed in all the Playa Poniente bivalve species. This

makes the average $\delta^{53}$Cr signature of a bivalve shell still a valuable parameter which, given that the

isotopic offset from ambient seawater is known, potentially can be used for recording the seawater Cr

isotope signature prevailing at the habitat location of the respective bivalve.

**5.7 A first attempt to define average $\delta^{53}$Cr offsets of specific bivalves from ambient seawater**


Our data set allow for a preliminary definition of Cr isotope offsets between certain

bivalve species and ambient seawater, which potentially could be used in paleo-seawater

reconstructions using suitable fossil aliquots. Instead of using average $\delta^{53}$Cr values defined by our

sample suites, and average seawater $\delta^{53}$Cr values, we prefer to define such offsets ($\Delta_{Cr}$) conservatively,

using band widths (rather than comparing average values) that take analytical uncertainty into

consideration (i.e., minimum $\Delta_{Cr}$ values defined by difference between $(\delta^{53}Cr + 2\sigma)_{sample}$ and $(\delta^{53}Cr - 2\sigma)_{seawater}$ ; maximum $\Delta_{Cr}$ values defined by difference between $(\delta^{53}Cr - 2\sigma)_{sample}$ and $(\delta^{53}Cr + 2\sigma)_{seawater}$). These ranges are listed in Table 3 and plotted in Figure 9 for all species where we have

multiple analyses and ambient seawater values. The $\Delta_{Cr}$ offset range of *Placuna placenta* is not strictly



comparable to the other values as it includes growth segment analyses covering the growth period of the entire shell. These introduce enhanced scatter that is most likely due to seasonal changes of seawater, a factor which is smoothed out by the analyses of entire shells as is the case for the other species. This explains the rather large $\Delta_{Cr}$ range calculated for *Placuna placenta*.

Although preliminary (additional data need to be collected to more precisely define

species dependent ranges), our data allow for a first order estimation on the use of the $\Delta_{Cr}$ seawater offset ranges defined herein to ultimately reconstruct the local surface seawater redox state. At average, $\delta^{53}$Cr values of ambient seawater can be reconstructed to ~+/- 0.3 ‰. At first sight, this seems to be rather imprecise, but considering that surface seawaters today exhibit $\delta^{53}$Cr variations between +0.13‰ and +1.24‰ (Paulukat et al., 2016), this uncertainty nevertheless allows for placing

reconstructed seawater compositions into a meaningful redox framework. The usefulness of this tool for the reconstruction of paleo-seawater compositional changes awaits the assessment, testing and acquisition of Cr isotope composition of fossil calcifiers that can be compared to data from modern respective species.

**6. Conclusions**

We have conducted bulk $\delta^{53}$Cr and [Cr] analyses of a set of common bivalve species from two locations, one at Playa Poniente on the Mediterranean Sea, and one from Disko Bay in arctic north Atlantic, from where we also measured the ambient seawater. Collection of same species during a

specific period in July over several years, and of multiple samples from some of the species, allowed us to monitor the stability of Cr isotope signatures in each of the species, and to define long-term $\delta^{53}$Cr offsets from ambient seawater. The outcome of our study can be summarized as follows:

1. The local surface seawater Cr isotope composition and [Cr] concentrations at Playa Poniente at times of sample collection over a three years period is surprisingly

homogenous, with $\delta^{53}$Cr = 0.83 +/- 0.05 ‰, and with [Cr] = 254 +/- 54 ng/kg.





2. Offsets ($\Delta_{Cr}$) from different bivalve species from this value show subtle differences, with typical values of ~0.3 to 0.4 ‰ lower than ambient seawater. Of all the species investigated, *Loripes lucinalis* exhibits the largest $\Delta_{Cr}$ of ~0.6 ‰. The systematically lighter Cr isotope compositions of all bivalves studies herein relative to ambient seawater confirms earlier studies by Pereira et al. (2015) on corals, by Wang et al. (2016) on foraminifera, and by Farkaš et al. (subm.) for various marine calcifiers from a location in the Barrier Reef.

3. Recognizing the importance of organics in the shell structures of bivalves, and considering our results from incinerated *vs.* solely 6N HCl dissolved bivalve shells systematically showing recovery of higher [Cr] in ashed samples, we propose a model whereby reduction of Cr(VI) originally contained in the seawater and transported to the calcifying space, to Cr(III), and its effective adsorption onto organic macromolecules that adhere to chitinous interlameallar coatings, plays a central role. In such a scenario, organic matter-bound, isotopically light Cr, forms preferable loci for the nucleation of carbonates, and it is eventually included into the growing shell carbonates, possible together with dissolved chromate that may substitute for $CO_3^{2-}$ directly in the carbonate lattice.

4. Inter-species Cr isotope variations, tested on a suite of contemporaneously sampled alive *Mytilus edulis* samples from Godthavn (Disko Bay), are small, in the range of $\delta^{53}Cr$ = +/- 0.05 ‰, and independent of [Cr]. Although not knowing the exact host of Cr in these shells (periostracum, organic macromolecules, chitinous interlamellar membranes ? etc.), the homogenous Cr isotope composition measured in this suite of samples renders *Mytilus edulis* a potential archive for the reconstruction of the redox state of ambient local seawater. This needs to be verified by studies of this species from other locations before attempts to use fossil aliquots for the reconstruction of paleo-seawater redox.



5. Intra-shell variations of $\delta^{53}$Cr and [Cr] over respective entire growth periods was investigated on two examples, a sample of *Placuna placenta* (window pane oyster, Capiz) and a sample of *Mimachlamys townsendi* (Pecinidae), from Kakinada Bay (Bay of Bengal) and from Hawke's Beach (Karachi, Pakistan). We observe subtle fluctuation of both parameters of the growth period of ~1 year and several years, respectively, which are in the order of 0.1 to 0.2 ‰. These fluctuations may arise from either seasonal changes in ambient seawater compositions, and/or from metabolic instabilities in the calcifying space affecting reduction of Cr(VI) and production of organic macromolecules.

6. Our study can be used as a base for futures, more detailed investigations of marine biogenic carbonates, including fossil marine calcifiers, aimed at the reconstruction of paleo-seawater redox state fluctuations, and eventually to correlate these with climate change aspects in certain periods of Earth's history.

*Author contribution*

RF initiated the study and collected the samples, RF and CP processed the samples through the chemistry and performed the mass spectrometrical analyses, and continuous discussions through the lengthy project period amongst all co-authors (RF, CP, SB and RK) led to substantial improvement, enhanced understanding, important modifications and adaptations of the original research ideas. RF prepared the manuscript with contributions from all co-authors.



*Acknowledgments*

We would like to thank Toby Leeper for always maintaining the mass spectrometers in perfect

running conditions and Toni Larsen for lab-assistance. Financial support through the Danish Agency for

Science, Technology and Innovation grant number 11-103378 to RF is highly appreciated.

.




**Figure captions**

Figure 1.

World sketch map with locations from where bivalve and seawater samples were collected.

Figure 2.

Photographs of representative bivalve species studied herein. 1-8 from Playa Poniente, 9 from

Kakinada Bay, 10 from Hawke's Bay, and 11 from Godhavn (Qeqertarsuaq). Black scale line

corresponds to 1 cm.

Cardiidae (unknown species); 2. *Pecten jacobaeus*; 3. *Challista chione*; 4. *Glycymeris glycymeris*; 5.

*Chamelea striulata*; 6. *Loripes lucinalis*, 7. *Venus verrucosa*; 8. *Arca navicularis*; 9. *Placuna placenta*

(window pane oyster, Capiz; with growth profile samples indicated); 10. *Mimachlamys townsendi*

(with growth profile samples indicated); 11. *Mytilus edulis*.


Figure 3.

Plot depicting average $\delta^{53}$Cr values from multiple filament runs with 200 nanogram loads of NIST SRM

979 measured on the PHOENIX thermal ionization mass spectrometer at 53Cr beam intensities of

350mV and 1V, respectively. The yellow colored range indicates the +/- 0.08 ‰ external

750     reproducibility of the 10 filaments loads ran at 350mV 53Cr beam intensities, which correspond to

typical beam intensities obtained from our samples.

Figure 4.

Plot showing the chromium concentrations ([Cr]; blue filled symbols) and chromium isotope

755     compositions ($\delta^{53}$Cr; red filled symbols) of various incinerated bivalve species analyzed from Playa

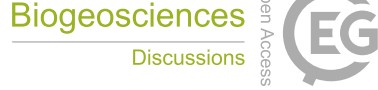

Poniente. 1 *Callista Chione*; 2 Cardiidae (species unknown); 3 *Chamelea gallina*; 4 *Chamelea striatula*;

5 *Glycymeris glycymeris*; 6 *Loripes lucinalis*; 7 *Pecten jacobaeus*; 8 *Venus verrucosa*. Stippled red lines

mark the average values of inter-species analyses, the red ranges the two standard deviation errors of

these analyses. The light gray horizontal bar depicts the Igneous Earth inventory composition

(Schoenberg et al., 2009), and the blue horizontal bar the local surface seawater composition

measured from this location. One sample of *Loripes lucinalis* (marked with a light red - and a light blue

filled symbol) has been dissolved in *aqua regia* without previous incineration. For details see text.

Figure 5.

Plot showing the chromium concentrations ([Cr]; blue filled symbols) and chromium isotope

compositions ($\delta^{53}$Cr; red filled symbols) of various *Mytilus edulis* shells and a of a shell mixture from

Godhavn, Disko Bay, Greenland. The darker red - and darker blue filled symbol mark analyses on

incinerated shells, whereas the lighter colored respective symbols depict analyses from solely *aqua*

*regia* dissolved shells. The light gray horizontal bar depicts the Igneous Earth inventory composition

(Schoenberg et al., 2009), and the blue horizontal bar the local surface seawater composition

measured from this location (Paulukat et al., 2016). For details see text.

Figure 6.

Plot showing the chromium concentrations ([Cr]; blue filled symbols) and chromium isotope

compositions ($\delta^{53}$Cr; red filled symbols) of samples along a growth transect of a *Placuna placenta*

sample (depicted in Fig. 2) from Kakinada Bay. Sample Cap-A is characterized by an elevated [Cr] which

is potentially due an elevated organic content in the initial growth zone comprising the apex of the

shell. The sinusoidal distribution of $\delta^{53}$Cr values along the transect potentially reflects seasonal

changes of ambient surface seawater. For details refer to text.


Figure 7.





Plot showing the chromium concentrations ([Cr]; blue filled symbols) and chromium isotope
compositions ($\delta^{53}$Cr; red filled symbols) of samples along a growth transect in a *Mimachlamys
townsendi* specimen from Hawke's Bay. Sample Pec-H is characterized by an elevated [Cr] which is
potentially due an elevated organic content in the initial growth zone comprising the apex/hinge of
the shell (c.f., Fig. 2). For details refer to text.

Figure 8.
Schematic representation of a simplified model for the transfer of chromium from an extrapallial fluid
within an interlamellar space into shell carbonate nuclides (modified from Suzuki and Nagasawa,
2013). The insoluble frameworks consist of chitin (black and long rectangles) that make a scaffold to
supply the space for precipitation of calcium carbonate crystals. A. The interlamellar space is filled
with a supersaturated extrapallial fluid with respect to $CO_3^{2-}$ and $Ca^{2+}$. Cr(VI) likely occurs as dissolved
compounds in the fluid and eventually is reduced to isotopically lighter Cr(III) by dissolved organic
macro-molecules (gray circles) onto which it is efficiently adsorbed. Insoluble matrix proteins (gray
discs) have the hydrophobic region for organic macro-molecule (protein)–chitin interaction and the
hydrophilic–acidic region for the calcium carbonate binding ability mediate the connection between
the organic scaffolds. B. The soluble matrix proteins that have hydrophyllic region for calcium
carbonate binding adhere to the chitinous layers and probably regulate nucleation, crystal polymorph
and crystal orientation of inorganic calcium carbonate crystals (gray rectangles). C. As the crystals
grow, insoluble matrix proteins are used for organic framework formation as intercrystalline organic
matrices and soluble matrix proteins, including adsorbed Cr(III), are eventually included in the calcium
carbonate crystals as intracrystalline organic matrices. Cr(VI) potentially present as chromate ($CrO_4^{2-}$)
ions likely also substitute for carbonate ($CO_3^{2-}$) ions directly in the calcium carbonate lattice (Tang et
al., 2009).



Figure 9. Bar graph showing the conservative offset ranges ($\Delta_{Cr}$) of $\delta^{53}$Cr values of bivalve species from

ambient seawater. The larger range of *Placuna placenta* is due to within-shell heterogeneities

probably resulting from seasonal surface seawater fluctuations which are smoothed out by the bulk

shell analyses of the other species (see text for details).





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





**Table 1. Chromium isotope compositons and chromium concentrations of surface seawaters**

| Sample | Cr [ng/kg] | ln[Cr] | $\delta^{53}$Cr [‰] | ±2σ | n | Year of Collection | Latitude/longitude | Reference |
|---|---|---|---|---|---|---|---|---|
| **Godhavn, Disko Bay, Greenland** | | | | | | | | |
| Disko Island 1 | 165 | 5.1 | 0.74 | 0.04 | 1 | 2016 | N 69°12′ W53°31′ | *Paulukat et al. (2016)* |
| Disko Island 2 | 185 | 5.2 | 0.70 | 0.03 | 1 | 2016 | | *Paulukat et al. (2016)* |
| Diske Island 3 | 179 | 5.2 | 0.75 | 0.10 | 2 | 2016 | | *Paulukat et al. (2016)* |
| | | average/2σ | **0.73** | **0.05** | | | | |
| *Playa Poniente, Benidorn, Spain* | | | | | | | | |
| Playa Poniente 1 | 280 | 5.6 | 0.86 | 0.07 | 4 | 2014 | N38°32′4.20″ W0°8′57.30″ | *Paulukat et al. (2016)* |
| Playa Poniente 2 | 271 | 5.6 | 0.82 | 0.08 | 2 | 2015 | | *this study* |
| Playa poniente 3 | 243 | 5.5 | 0.85 | 0.07 | 2 | 2016 | | *this study* |
| Playa Poniente 4 | 222 | 5.4 | 0.81 | 0.07 | 3 | 2017 | | *this study* |
| | | average/2σ | **0.83** | **0.05** | | | | |
| Playa Albir | 239 | 5.5 | 0.90 | 0.17 | 3 | 2013 | N38°34′36.72″ W0° 3′46.56″ | *Paulukat et al. (2016)* |
| Playa Albir | 306 | 5.7 | 0.81 | 0.03 | 1 | 2014 | | *Paulukat et al. (2016)* |
| Playa Albir | 301 | 5.7 | 0.96 | 0.02 | 1 | 2015 | | *Paulukat et al. (2016)* |
| | | average/2σ | **0.89** | **0.15** | | | | |



**Table 2. Chromium isotope compositons and chromium concentrations of bivalves**

| Sample | | Family / Species | Cr [ppm] | $\delta^{53}$Cr [‰] | ±2σ | n | $D_{Cr}$ | $\Delta_{Cr}$ (band width; ‰) | Year of collection |
|---|---|---|---|---|---|---|---|---|---|
| **Godhavn, Disko Bay, Greenland** | | | | | | | | | |
| God-1 | | *Mytilus edulis* (Mytilidae) | 0.045 | 0.09 | 0.06 | 6 | 256 | | 2016 |
| God-2 | | *Mytilus edulis* (Mytilidae) | 0.041 | 0.10 | 0.09 | 1 | 233 | | 2016 |
| God-3 | | *Mytilus edulis* (Mytilidae) | 0.032 | 0.15 | 0.06 | 4 | 182 | | 2016 |
| God-4 | | *Mytilus edulis* (Mytilidae) | 0.037 | 0.11 | 0.07 | 3 | 210 | | 2016 |
| God-4 ashed | | *Mytilus edulis* (Mytilidae) | 0.070 | 0.12 | 0.08 | 2 | 398 | | 2016 |
| God-5 | | *Mytilus edulis* (Mytilidae) | 0.043 | 0.13 | 0.08 | 2 | 244 | | 2016 |
| God-5 ashed | | *Mytilus edulis* (Mytilidae) | 0.068 | 0.14 | 0.07 | 3 | 386 | | 2016 |
| God 6 | | *Mytilus edulis* (Mytilidae) | 0.037 | 0.09 | 0.05 | 5 | 210 | | 2016 |
| God mix | | *Mytilus edulis* (Mytilidae) | 0.041 | 0.12 | 0.07 | 5 | 233 | | 2016 |
| God mix ashed | | *Mytilus edulis* (Mytilidae) | 0.066 | 0.08 | 0.07 | 5 | 375 | | 2016 |
| | | average | | 0.11 | | | 273 | from 0.52 to 0.72 | |
| | | 2 σ | | 0.05 | | | 162 | | |
| **Hawke's Bay Beach, Karachi, Pakistan** | | | | | | | | | |
| Pec-A | margin | *Mimachlamys townsendi* (Pectenidae) | 0.033 | 0.06 | 0.04 | 3 | 110 | | 1969 |
| Pec-B | | *Mimachlamys townsendi* (Pectenidae) | 0.021 | 0.09 | 0.07 | 4 | 70 | | 1969 |
| Pec-C | | *Mimachlamys townsendi* (Pectenidae) | 0.063 | 0.01 | 0.07 | 7 | 210 | | 1969 |
| Pec-D | | *Mimachlamys townsendi* (Pectenidae) | 0.031 | 0.16 | 0.08 | 4 | 103 | | 1969 |
| Pec-E | | *Mimachlamys townsendi* (Pectenidae) | 0.045 | 0.08 | 0.06 | 5 | 150 | | 1969 |
| Pec-F | | *Mimachlamys townsendi* (Pectenidae) | 0.029 | 0.05 | 0.05 | 4 | 97 | | 1969 |
| Pec-G | | *Mimachlamys townsendi* (Pectenidae) | 0.037 | 0.13 | 0.05 | 4 | 123 | | 1969 |
| Pec-H | hinge | *Mimachlamys townsendi* (Pectenidae) | 0.389 | 0.01 | 0.04 | 8 | 1297 | | 1969 |
| | | average | | 0.07 | | | 270 | - | |
| | | 2 σ | | 0.11 | | | 834 | | |
| **Katinaga Bay, Andhra Pradesh, India** | | | | | | | | | |
| Cap-A | hinge | *Placuna placenta (Capiz shell, window pane oyster)* | 0.246 | -0.01 | 0.06 | 8 | 820 | | 1969 |
| Cap-B | | *Placuna placenta (Capiz shell, window pane oyster)* | 0.057 | 0.08 | 0.07 | 8 | 190 | | 1969 |
| Cap-C | | *Placuna placenta (Capiz shell, window pane oyster)* | 0.061 | 0.18 | 0.04 | 4 | 203 | | 1969 |
| Cap-D | | *Placuna placenta (Capiz shell, window pane oyster)* | 0.137 | 0.13 | 0.07 | 3 | 457 | | 1969 |
| Cap-E | | *Placuna placenta (Capiz shell, window pane oyster)* | 0.077 | -0.08 | 0.04 | 3 | 257 | | 1969 |
| Cap-F | | *Placuna placenta (Capiz shell, window pane oyster)* | 0.057 | 0.06 | 0.08 | 2 | 190 | | 1969 |
| Cap-G | margin | *Placuna placenta (Capiz shell, window pane oyster)* | 0.030 | -0.04 | 0.08 | 3 | 100 | | 1969 |
| | | average | | 0.05 | | | 317 | from 0.23 to 0.77 | |
| | | 2 σ | | 0.19 | | | 496 | | |
| ***Playa Poniente, Benidorn, Spain*** | | | | | | | | | |
| PP15-J | | *Arca Navicularis* | 0.191 | 0.570 | 0.11 | 4 | 752 | | 2015 |
| PPS-02 | | *Arca Navicularis* | 0.052 | 0.166 | 0.04 | 1 | 204 | | 2015 |
| PP15-A | | *Callista Chione* | 0.066 | 0.461 | 0.07 | 5 | 260 | | 2015 |
| PP15-E | | *Callista Chione* | 0.040 | 0.422 | 0.10 | 2 | 157 | | 2015 |
| PP15-G | | *Callista Chione* | 0.062 | 0.345 | 0.09 | 1 | 244 | | 2015 |
| PP15-H | | *Callista Chione* | 0.051 | 0.387 | 0.09 | 1 | 201 | | 2015 |
| PP15-I | | *Callista Chione* | 0.073 | 0.327 | 0.10 | 1 | 287 | | 2015 |
| PPS-09 (1) | | *Callista Chione* | 0.060 | 0.409 | 0.03 | 1 | 236 | | 2015 |
| PPS-09 (2) | | *Callista Chione* | 0.070 | 0.316 | 0.13 | 2 | 276 | | 2015 |
| PP16 16 | | *Callista Chione* | 0.031 | 0.296 | 0.09 | 1 | 122 | | 2016 |
| PP16 9 | | *Callista Chione* | 0.068 | 0.468 | 0.09 | 4 | 268 | | 2016 |
| PP16 10 | | *Callista Chione* | 0.079 | 0.490 | 0.06 | 5 | 311 | | 2016 |
| PP16 11 | | *Callista Chione* | 0.055 | 0.359 | 0.09 | 2 | 217 | | 2016 |
| PP17-11 | | *Callista Chione* | 0.070 | 0.412 | 0.08 | 2 | 274 | | 2017 |
| PP17-21 | | *Callista Chione* | 0.031 | 0.336 | 0.05 | 2 | 122 | | 2017 |
| PP17-22 | | *Callista Chione* | 0.038 | 0.326 | 0.05 | 3 | 150 | | 2017 |
| PP17-25 | | *Callista Chione* | 0.034 | 0.464 | 0.09 | 3 | 134 | | 2017 |
| PP17-29 | | *Callista Chione* | 0.068 | 0.464 | 0.04 | 5 | 268 | | 2017 |
| PP17-30 | | *Callista Chione* | 0.029 | 0.480 | 0.01 | 2 | 114 | | 2017 |
| PP17-31 | | *Callista Chione* | 0.069 | 0.493 | 0.03 | 5 | 272 | | 2017 |
| | | | | | | | | from 0.25 to 0.61 | |
| PPS-03 (1) | | *? (Cardiidae)* | 0.059 | 0.667 | 0.02 | 1 | 230 | | 2015 |
| PP16 13 | | *? (Cardiidae)* | 0.052 | 0.520 | 0.07 | 4 | 205 | | 2016 |
| PP16 15 | | *? (Cardiidae)* | 0.049 | 0.590 | 0.10 | 4 | 193 | | 2016 |
| PP17-10 | | *? (Cardiidae)* | 0.044 | 0.636 | 0.09 | 2 | 175 | | 2017 |
| | | | | | | | | from 0.05 to 0.41 | |
| PP16 14 | | *Chamelea gallina* | 0.049 | 0.310 | 0.08 | 4 | 193 | - | 2016 |
| White shell (1) | | *Chamelea striatula* | 0.094 | 0.450 | 0.11 | 2 | 370 | | 2014 |
| White shell (2) | | *Chamelea striatula* | 0.072 | 0.560 | 0.12 | 1 | 283 | | 2014 |
| PP15-C | | *Chamelea striatula* | 0.078 | 0.577 | 0.09 | 5 | 307 | | 2015 |
| PPS-07 (1) | | *Chamelea striatula* | 0.047 | 0.463 | 0.05 | 1 | 185 | | 2015 |



**Table 2. Chromium isotope compositons and chromium concentrations of bivalves**

| Sample | Family / Species | Cr [ppm] | δ⁵³Cr [‰] | ±2σ | n | $D_{Cr}$ | Δ_Cr (band width; ‰) | Year of collection |
|---|---|---|---|---|---|---|---|---|
| PPS-07 (2) | *Chamelea striatula* | 0.060 | 0.543 | 0.09 | 2 | 236 | | 2015 |
| PPS-07 (3) | *Chamelea striatula* | 0.070 | 0.590 | 0.09 | 3 | 276 | | 2015 |
| PP16 17 | *Chamelea striatula* | 0.047 | 0.390 | 0.06 | 3 | 185 | | 2016 |
| PP17-8 | *Chamelea striatula* | 0.094 | 0.373 | 0.07 | 3 | 371 | | 2017 |
| PP17-9 | *Chamelea striatula* | 0.093 | 0.401 | 0.07 | 3 | 366 | | 2017 |
| PP17-17 | *Chamelea striatula* | 0.081 | 0.537 | 0.07 | 5 | 319 | | 2017 |
| PP17-28 | *Chamelea striatula* | 0.061 | 0.464 | 0.12 | 4 | 240 | | 2017 |
| | | | | | | | *from 0.13 to 0.55* | |
| PP15-B | Glycymeris | 0.084 | 0.358 | 0.08 | 5 | 331 | | 2015 |
| PP15-F | Glycymeris | 0.042 | 0.359 | 0.08 | 4 | 165 | | 2015 |
| PPS-06 (1) | Glycymeris | 0.100 | 0.521 | 0.11 | 2 | 394 | | 2015 |
| PPS-06 (2) | Glycymeris | 0.060 | 0.533 | 0.05 | 2 | 236 | | 2015 |
| PPS-06 (3) | Glycymeris | 0.060 | 0.415 | 0.08 | 3 | 236 | | 2015 |
| PP16 6 | Glycymeris | 0.093 | 0.440 | 0.08 | 6 | 366 | | 2016 |
| PP16 19 | Glycymeris | 0.050 | 0.397 | 0.06 | 5 | 197 | | 2016 |
| PP17-5 | Glycymeris | 0.060 | 0.452 | 0.05 | 2 | 235 | | 2017 |
| PP17-13 | Glycymeris | 0.081 | 0.452 | 0.08 | 3 | 318 | | 2017 |
| PP17-2 | Glycymeris | 0.073 | 0.487 | 0.08 | 3 | 289 | | 2017 |
| PP17-3 | Glycymeris | 0.095 | 0.600 | 0.04 | 5 | 376 | | 2017 |
| PP17-7 | Glycymeris | 0.111 | 0.552 | 0.08 | 4 | 435 | | 2017 |
| PP17-12 | Glycymeris | 0.091 | 0.574 | 0.07 | 3 | 359 | | 2017 |
| | | | | | | | *from 0.15 to 0.57* | |
| White shell (3) | *Loripes lucinalis* | 0.047 | 0.250 | 0.12 | 1 | 185 | | 2014 |
| White shell (4) | *Loripes lucinalis* | 0.045 | 0.200 | 0.09 | 1 | 177 | | 2014 |
| White shell (3) | *Loripes lucinalis* | 0.046 | 0.250 | 0.12 | 1 | 181 | | 2014 |
| White shell (4) | *Loripes lucinalis* | 0.046 | 0.200 | 0.09 | 1 | 181 | | 2014 |
| PP15-D | *Loripes lucinalis* | 0.053 | 0.312 | 0.07 | 3 | 209 | | 2015 |
| PP15-D unashed | *Loripes lucinalis* | 0.027 | 0.286 | 0.07 | 3 | 106 | | 2015 |
| PPS-08 (1) | *Loripes lucinalis* | 0.070 | 0.159 | 0.14 | 2 | 276 | | 2015 |
| PPS-08 (2) | *Loripes lucinalis* | 0.060 | 0.157 | 0.18 | 3 | 236 | | 2015 |
| PPS-08 (3) | *Loripes lucinalis* | 0.053 | 0.170 | 0.10 | 2 | 209 | | 2015 |
| PP16 12 | *Loripes lucinalis* | 0.043 | 0.199 | 0.06 | 4 | 169 | | 2016 |
| PP16 12 | *Loripes lucinalis* | 0.044 | 0.200 | 0.06 | 4 | 173 | | 2016 |
| PP17-16 | *Loripes lucinalis* | 0.040 | 0.162 | 0.04 | 5 | 157 | | 2017 |
| PP17-6 | *Loripes lucinalis* | 0.044 | 0.253 | 0.06 | 2 | 172 | | 2017 |
| PP17-16 | *Loripes lucinalis* | 0.038 | 0.265 | 0.05 | 3 | 150 | | 2017 |
| PP17-14 | *Loripes lucinalis* | 0.040 | 0.170 | 0.04 | 5 | 157 | | 2017 |
| PP17-15 | *Loripes lucinalis* | 0.043 | 0.194 | 0.08 | 3 | 169 | | 2017 |
| | | | | | | | *from 0.47 to 0.77* | |
| PPS-04 (1) | *Pecten jacobaeus* | 0.127 | 0.461 | 0.08 | 2 | 500 | | 2015 |
| PPS-04 (2) | *Pecten jacobaeus* | 0.163 | 0.474 | 0.08 | 5 | 640 | | 2015 |
| PP17-1 | *Pecten jacobaeus* | 0.151 | 0.502 | 0.09 | 5 | 593 | | 2017 |
| | | | | | | | *from 0.26 to 0.44* | |
| PP17-18 | *Venus verrucosa* | 0.090 | 0.324 | 0.16 | 3 | 354 | | 2017 |
| PP17-32 | *Venus verrucosa* | 0.058 | 0.340 | 0.05 | 2 | 228 | | 2017 |
| | | | | | | | *from 0.43 to 0.57* | |
| PPS-05 | *Venus nux* | 0.090 | 0.467 | 0.14 | 4 | 354 | | 2015 |



# Figure 1



# Figure 2

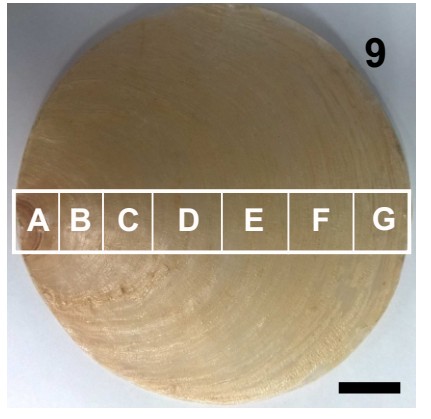

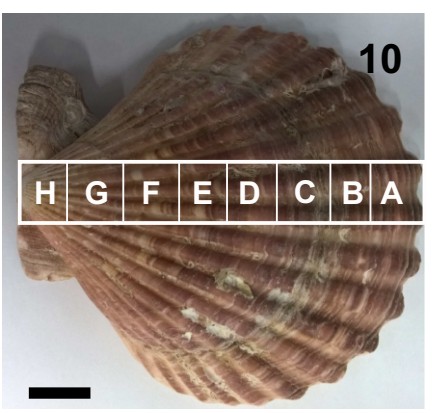

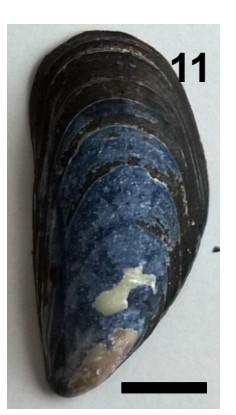





# Figure 3



# Figure 4





Figure 5





# Figure 6







# Figure 7



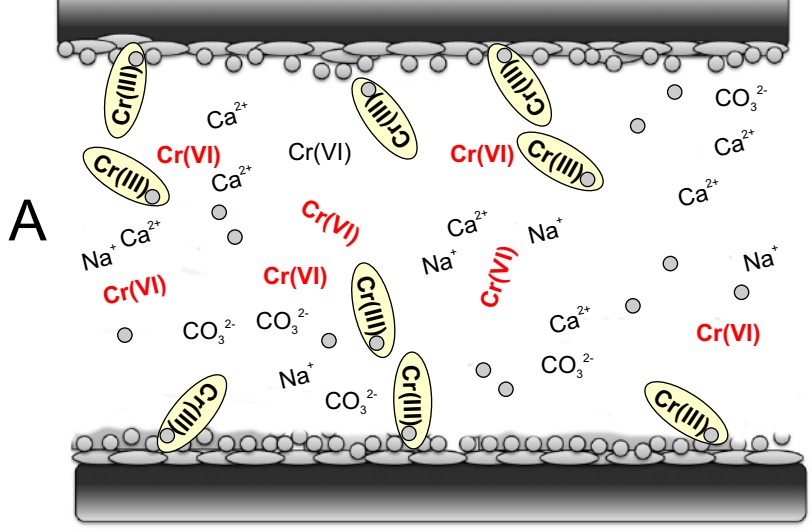

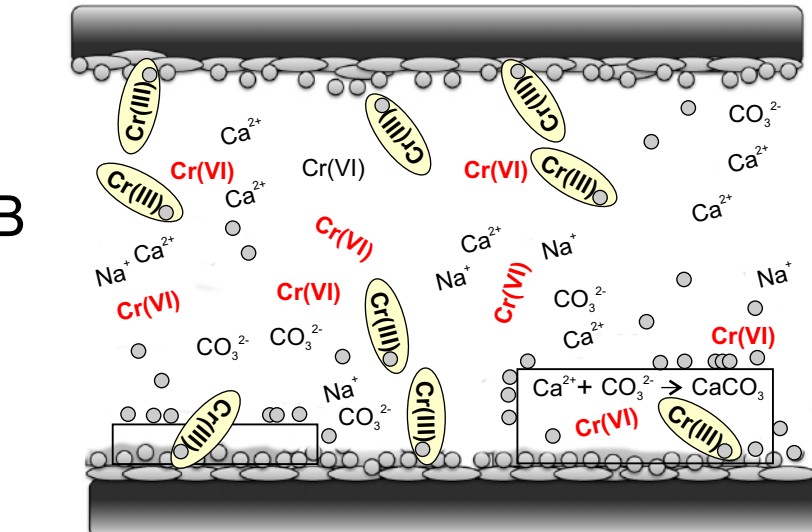

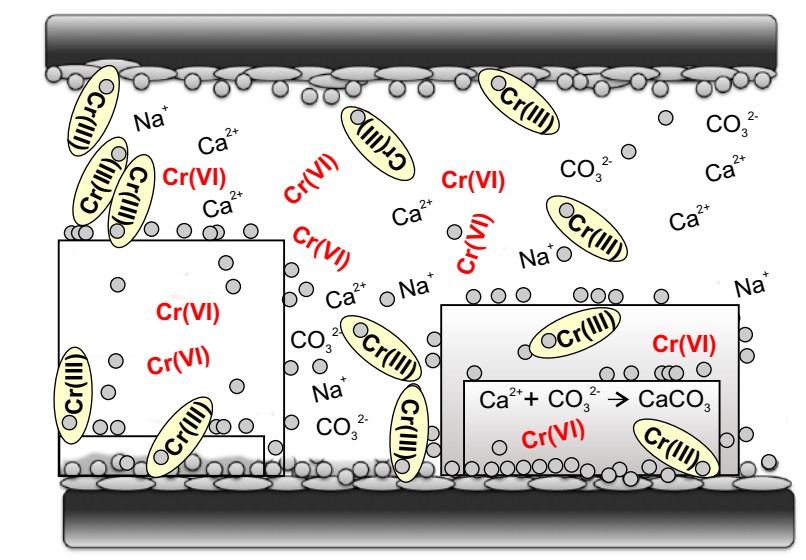

Figure 8





Figure 9

