# Peer review of "A systematic look at chromium isotopes in modern shells – implications for paleoenvironmental reconstructions"

_Biogeosciences, 2018_

## Referee Comment (RC1) · Anonymous Referee #1 · 14 May 2018

Comments to the authors

The authors provide new data of Cr isotope composition in biogenic carbonate (shells) and add further information regarding the potential of the $\delta53Cr$ as a proxy for paleo-seawater $\delta53Cr$ composition, and/or the past redox conditions of the sea water. The authors compared the $\delta53Cr$ signal of fast growing shells to local sea water and found a large offset between shells and sea water, similar to other biogenic carbonates (corals and foraminifera). The authors also provided a model explaining how calcification processes may generate the offset between biogenic carbonate and sea water. This study provides an important message to the biogeochemical community, especially those

interested in chromium stable isotopic applications. I think the manuscript is a nice contribution and deserves to be considered for publication in Biogeosciences.

Technical corrections 1. Please separate the unit from the number, there are many situation through the text that this is not followed. (e.g. Line 137 "5m"; line 150 "8mm") 2. Line 321 – "d53Cr" correct to $\delta$53Cr 3. Lines 342, 349, 473 – idem 4. Line 382, "(Pereira et al., 2015)" correct to Pereira et al (2015) 5. Line 741, insert "1." at Cardiidae. 6. Line 766, exclude "a (yellow highlighted)" at the sentence "of various Mytilus edulis shells and a of a shell mixture"

---

## Author Comment (AC1) · 14 May 2018

Dear reviewer 1,

thank you for your positive response and comments to our manuscript. I will be dealing with the minor suggestions for corrections in the text once I have the official comment of the other reviewr(s), and come up with a final text with corrections implemented. With best regards and greetings Robert Frei
* * *

---

## Referee Comment (RC2) · Anonymous Referee #2 · 4 Jun 2018

Frei et al present new bivalve d53cr data accompanied by coexisting seawater d53cr data. They confirm that marine calcifiers produce carbonates with much lower d53cr compared to contemporaneous seawater. They also found large inter-species and small intra-species d53cr variations. The large proportion of Cr being hosted in bivalve shell organic matter is very interesting as well. Clearly, this work is a nice step forward developing bivalve as potential seawater d53cr archive, which can potentially help build a more detailed ocean-atmosphere redox record over the Phanerozoic Eon. Based on this, I recommend publication of this work in Biogeosciences. The following minor suggestions should be fairly easy to address.

[Figure]

Technical comments Currently there is significant amount of discussion mingled with results in the result section. I suggest separate out discussion from results.

When discussing distribution coefficients, it would be good to compare biogenic against abiogenic values, to emphasize the importance of biological processes.

Since currently there is still a lack of concrete evidence for Cr(VI) reduction during biological calcification, it may be good idea to equally discuss alternative hypotheses, such as uptaking organic-complexed Cr(III) from seawater.

371 Missing a period.

650. On average.

706. Future, not futures.

---

## Author Comment (AC2) · 6 Jun 2018

In the name of all co-authors I thank referee 2 for the positive feedback and suggestions for minor revisions. I will cope with all af them in the final revised manuscript in the way listed below:

Technical comments Currently there is significant amount of discussion mingled with results in the result section. I suggest separate out discussion from results.

Answer: I will remove parts of a discussion nature out of the resutls section and incorporate them where they belong to.

[Figure]

When discussing distribution coefficients, it would be good to compare biogenic against abiogenic values, to emphasize the importance of biological processes.

Answer: I will try to compute distributions factors of Cr into "abiogenic" limestones as to compare them with the distribution factors we report for biogenic carbonates in our manuscript.

Since currently there is still a lack of concrete evidence for Cr(VI) reduction during biological calcification, it may be good idea to equally discuss alternative hypotheses, such as uptaking organic-complexed Cr(III) from seawater.

Answer: Although there is no concrete data on the uptake of Cr via organic-complexed ligands, I will mention the reductive stripping of Cr(VI) via adsorption onto phytoplancton (Semeniuk et al., 2016) as one example in which Cr is reductivley removed by organic material.

371 Missing a period.

Answer: Will of course correct this.

650. On average.

Answer: will of course correct this.

706. Future, not futures.

Answer: will of course correct this.

---

## Author Response (AR1)

**Comments to reviews of our manuscript, # bg-2018-138**

Dear editor,

we appreciate the positive evaluation of our manuscript entitled "A systematic look at chromium isotopes in modern shells – implications for paleo-environmental reconstructions" by two anonymous reviewers. We are thankful for their constructive comments and the suggestions added by associate editor Aninda Mazumdar. We have implemented their suggestions in a revised manuscript and respond in a point-by-point manner below:

Referee #1

We have implemented and corrected the technical corrections pointed out by this reviewer.

1. All units are now separated from the respective numbers
2. Corrected to $\delta^{53}Cr$
3. Corrected to $\delta^{53}Cr$
4. Corrected to Pereira et al. (2015)
5. "1" inserted
6. Excluded "a"

Referee #2

Technical comments: we have now transferred discussion text inter mingled the results chapter to the actual discussion chapter. We now instead discuss the seawater data and data from the respective shell transect in separate short sub-chapters (5.1 and 5.2).

There is a lack of modern seawater – carbonate sediment pair data that would make calculations of distribution coefficients for abiogenic carbonates possible. We have added, at the end of chapter 5.3, a respective distribution coefficient value from the data in Holmden et al. (2016) for seawater and carbonate sediment form Jamaica, to our knowledge the only data set that allows for calculation of an abiogenic distribution coefficient.

To our recollection we intensely discuss all the possibilities for explaining the isotopically light Cr uptake into calcifying organisms in our discussion. We here emphasize that the isotopically light Cr signatures measured in all shells are compatible with results from foraminifera and corrals and all in all point to a redox process during Cr uptake into the shell carbonates. With the presently scarce data sets available, Cr(III) uptake via organic ligands would rather prefer isotopically heavy Cr as suggested by Saad et al. (2017), a contribution which we cite in our manuscript.

Line 371. Added the "."

Line 650: changed to "on average"

Line 750: changed to "future".

Associate editor Aninda Mazumdar

A.  We have clarified the requirements for sample sizes in the respective chapter. We do not have TOC values of the sample studied that could substantiate the relation between Cr concentrations and organic material in the shells. Our incineration experiments however clearly point to the importance of organic matter in this respect.

B.  We would like to keep the reference to the Farkas et al. paper as our study is a direct follow up on Farkas et al.'s investigations. The Farkas et al. manuscript is now in press with *Earth and Planetary Science Letters*, and it will be potentially published completely at the time our study could be published in *Biogeosciences*. For your convenience, we have uploaded the "proofs" of the Farkas et al. paper soon to appear. The reason why we would like to keep referring to this study is that it is the only one to data that allow a comparison of marine mollusk Cr isotope data with ambient seawater and therefore sets an appropriate frame for our contribution. We allows us to amend (following the our own marked up revised manuscript) the proofs of the Farkas et al paper as a guarantee/proof that this article is now officially in press and soon to be allocated respective volume, issue and page numbers.

C.  We fully agree that the Ce anomaly – Cr isotope relationship in Farkas et al. has no relationship to our study, and we have therefore deleted all the parts that refer to this peculiarity in our revised manuscript.

D.  It is correct that we do not have Eh and oxygen data on the surface waters in which the mollusks thrived. In this respect, we do not have control on seasonal variations in Cr concentrations and Cr isotopes that could substantiate the intra-species variations we see in the transect samples. All we can say at his stage is that we have results that show redox variations in surface waters recorded in Baltic seawater (study by Paulukat et al. 2015, cited in our manuscript), which clearly are dictated by algae blooms that fractionate and strip Cr effectively form the water columns.

E.  We would strongly disagree with this as hitherto no solid experimental data exist that constrains isotope effects during inorganic incorporation of Cr into carbonates. The only study that approaches this issue is that by Rodler et al. which we extensively cite in our manuscript.

F.  We have tried to shorten the manuscript as much as possible, particularly we have deleted sections in the chapter that describes our model in relation to known observations of organic material in shell structures. Also, the many referring paragraphs to Farkas et al. with respect to Ce anomalies are deleted. However, we would like to keep the aspects of paleo redox reconstructions that potentially are doable when appropriate data on modern systems are available. The major aim with using shells as marine Cr isotope archives is their potential use for reconstructing redox variations in ancient oceans, with important implications and links to climate change on land.

We hope that our revised manuscript is a significantly improved version that meets the standards of Biogeosciences for publication.

Thanks in advance and very best regards

Robert Frei (on behalf of all coauthors)

[revised manuscript text omitted]

**AUTHOR QUERY FORM**

| | **Journal:** EPSL | **Please e-mail your responses and any corrections to:** |
|---|---|---|
| ELSEVIER | **Article Number:** 15135 | **E-mail:** corrections.eseo@elsevier.vtex.lt |

Dear Author,

Please check your proof carefully and mark all corrections at the appropriate place in the proof. **It is crucial that you NOT make direct edits to the PDF using the editing tools as doing so could lead us to overlook your desired changes.** Rather, please request corrections by using the tools in the Comment pane to annotate the PDF and call out the changes you would like to see. To ensure fast publication of your paper please return your corrections within 48 hours.

For correction or revision of any artwork, please consult *http://www.elsevier.com/artworkinstructions*

Any queries or remarks that have arisen during the processing of your manuscript are listed below and highlighted by flags in the proof.

| Location in article | Query / Remark: **Click on the Q link** to find the query's location in text
Please insert your reply or correction at the corresponding line in the proof |
|---|---|
| Q1 | Your article is registered as a regular item and is being processed for inclusion in a regular issue of the journal. If this is NOT correct and your article belongs to a Special Issue/Collection please contact <f.kay@elsevier.com> immediately prior to returning your corrections. (p. 1/ line 1) |
| Q2 | The author names have been tagged as given names and surnames (surnames are highlighted in teal color). Please confirm if they have been identified correctly and are presented in the desired order. (p. 1/ line 17) |
| Q3 | Please indicate which author(s) should be marked as 'Corresponding author'. (p. 1/ line 18) |
| Q4 | The number of keywords provided exceeds the maximum of 6 allowed by this journal. Please provide the final list of 6 keywords for the article. (p. 1/ line 37) |
| Q5 | Please check the e-mail address that has been added here, and correct if necessary. (p. 1/ line 62) |
| Q6 | Figure(s) will appear in black and white in print and in color on the web. The figure(s) contains references to color or the colors are mentioned in the main text. Based on this, the explanatory text about the interpretation of the colors has been added. Please check, and correct if necessary. (p. 3/ line 50) |
| Q7 | Please check if sponsor names have been identified correctly and correct if necessary. (p. 14/ line 5) |
| Q8 | Please provide a grant number for this sponsor - University of Adelaide - (if available) and include this number into the main text where appropriate. (p. 14/ line 6) |

Please check this box or indicate
your approval if you have no
corrections to make to the PDF file

Thank you for your assistance.                                                                                      Page 1 of 1

Contents lists available at ScienceDirect

**Earth and Planetary Science Letters**

www.elsevier.com/locate/epsl

ELSEVIER

[Figure]

**Chromium isotope fractionation between modern seawater and biogenic carbonates from the Great Barrier Reef, Australia: Implications for the paleo-seawater $\delta^{53}$Cr reconstruction**

Juraj Farkaš [a,b,c], Jiří Frýda [d], Cora Paulukat [e,1], Edmund Hathorne [f], Šarka Matoušková [g], Jan Rohovec [g], Barbora Frýdová [d], Michaela Francová [a], Robert Frei [e]

[a] Department of Geochemistry, Czech Geological Survey, Prague, Czech Republic
[b] Department of Earth Sciences, University of Adelaide, North Terrace, Adelaide, Australia
[c] TRaX – Centre for Tectonics, Resources and Exploration, University of Adelaide, North Terrace, Adelaide, SA 5005, Australia
[d] Department of Environmental Geosciences, Czech University of Life Sciences, Prague, Czech Republic
[e] Department of Geosciences and Natural Resource Management, University of Copenhagen, Denmark
[f] GEOMAR, Helmholtz Centre for Ocean Research, Kiel, Germany
[g] Institute of Geology, Academy of Sciences of the Czech Republic, Czech Republic

**ARTICLE INFO**

*Article history:*
Received 18 January 2018
Received in revised form 22 June 2018
Accepted 22 June 2018
Available online xxxx
Editor: D. Vance

*Keywords:*
chromium
isotopes
REE
carbonates
seawater
Great Barrier Reef
redox

**ABSTRACT**

This study investigates chromium isotope variations ($\delta^{53}$Cr) and REE patterns in present-day biogenic carbonates and ocean waters from Lady Elliot Island (LEI) located in the southern Great Barrier Reef (GBR), Australia, which is one of the world's largest carbonate-producing shelf ecosystems. Our results from thoroughly cleaned biogenic carbonates collected at LEI, with no detectable evidence for lithogenic Cr and/or Mn–Fe oxide coating contamination, revealed a systematic and statistically significant correlation ($r^2 = 0.83$, $p < 0.05$) between $\delta^{53}$Cr and cerium anomaly (Ce/Ce*) data in molluscan shells (i.e., gastropods). This in turn implies a redox-controlled incorporation of Cr from seawater into a shell during mineralization mediated by the organism. In particular, shells with higher $\delta^{53}$Cr values, which approach the Cr isotope composition of local seawater, tend to be associated with more negative Ce/Ce*. Importantly, the intercept of the above $\delta^{53}$Cr vs. Ce/Ce* correlation points to the Cr isotope composition of local ocean water, which has an average $\delta^{53}$Cr of $+0.82 \pm 0.13‰$ ($2\sigma$, relative to SRM 979). These findings thus indicate that the above multi-proxy approach could be used to reconstruct the $\delta^{53}$Cr signature of local paleo-seawater based on Ce/Ce* and $\delta^{53}$Cr data in a set of well-preserved fossil skeletal carbonates (i.e., molluscan shells) collected at a specific site. Interestingly, the only calcifying organism from LEI that yielded identical $\delta^{53}$Cr vs. Ce/Ce* values as those in ambient ocean water was a microbial calcitic carbonate produced by red coralline algae (*Lithothamnion* sp.). This organism thus seems to incorporate Cr isotopes and REE from seawater without additional biological discrimination and/or isotope fractionation effects. Considering that calcite is a more stable CaCO₃ polymorph during post-depositional alternation and diagenetic stabilization of marine carbonates (compared to aragonite), the fossil counterparts of these algal-microbial carbonates (microbialites) might thus represent ideal natural archives of the paleo-seawater $\delta^{53}$Cr and Ce/Ce* variations over geological time.

Finally, our compilation of $\delta^{53}$Cr data from recent marine biogenic carbonates originating from the main oceanic provinces (South/North Pacific, South/North Atlantic, Caribbean, Mediterranean Sea) confirms that marine carbonates tend to be systematically enriched in light Cr isotopes relative to local ocean waters. Trace element constraints, however, indicate that some of these shifts to lower $\delta^{53}$Cr values (i.e., approaching −0.1 per mil) are related to a presence of lithogenic Cr in the shells, causing a diagenetic overprint of the primary marine $\delta^{53}$Cr signal.

© 2018 Elsevier B.V. All rights reserved.

**1. Introduction**

Chromium (Cr) is a redox-sensitive transition metal with four naturally occurring isotopes ($^{50}$Cr, $^{52}$Cr, $^{53}$Cr and $^{54}$Cr) that could

*E-mail address:* juraj.farkas@adelaide.edu.au (J. Farkaš).
1 Current address: ALS Scandinavia AB, Aurorum 10, 977 75, Luleå, Sweden.

https://doi.org/10.1016/j.epsl.2018.06.032

0012-821X/© 2018 Elsevier B.V. All rights reserved.

be used as tracers to constrain present and past redox conditions on Earth and other solar system objects (Bonnand et al., 2016; Schoenberg et al., 2016; Crowe et al., 2013; Planavsky et al., 2014; Frei et al., 2009). Stable Cr isotopes can also be utilized to monitor the sources and biogeochemical pathways of chromium within terrestrial reservoirs including geological, hydrological and biological systems (Paulukat et al., 2016; Holmden et al., 2016; D'Arcy et al., 2016; Wang et al., 2016; Pereira et al., 2015; Scheiderich et al., 2015; Farkaš et al., 2013; Bonnand et al., 2013; Schoenberg et al., 2008; Ellis et al., 2002). Although Cr can occur in numerous oxidation states (ranging from −2 to +6; cf., Daulton and Little, 2006), in most near-surface terrestrial environments including the oceans, it is typically present either as trivalent Cr(III) or hexavalent Cr(VI) species, depending on local redox conditions (Kotaś and Stasicka, 2000; Elderfield, 1970; Pettine and Millero, 1990).

Importantly, there is a systematic fractionation of stable Cr isotopes (i.e., $^{53}Cr/^{52}Cr$ ratios or $\delta^{53}Cr$) in nature due to redox processes, where the reduction of Cr in near-surface environments produces dissolved Cr(VI) that is isotopically heavier, relative to a less soluble Cr(III) species (Ellis et al., 2002; Zink et al., 2010; Døssing et al., 2011). The Cr isotope composition of seawater thus reflects a complex signal of oxidation/reduction processes operating within the oceans (Scheiderich et al., 2015; Paulukat et al., 2016), and the $\delta^{53}Cr$ record of marine sedimentary archives has potential to be used to infer the past redox conditions of the ocean–atmosphere system through geological time (Frei et al., 2009, 2011, 2013, 2016; Bonnand et al., 2013; Van Zuilen and Schoenberg, 2013; Planavsky et al., 2014; Holmden et al., 2016; D'Arcy et al., 2016; Rodler et al., 2016; Gilleaudeau et al., 2016).

Due to a basically continuous geological record of marine carbonates throughout most of the Earth's history, i.e., the last ∼3.7 billion years (Nutman et al., 2016; Shields and Veizer, 2002), Cr isotope studies of marine carbonate archives are particularly appealing for paleo-redox reconstructions. However, as shown by recent studies (Rodler et al., 2015; Pereira et al., 2015; Wang et al., 2016), the actual mechanism(s) of Cr incorporation and redox-controlled isotope fractionation during the formation of inorganic and biogenic carbonates is rather complex and poorly understood, thus requiring further systematic investigations in both natural and laboratory-controlled settings. Available results from inorganic calcite precipitation experiments have revealed that the incorporation of Cr from a solution into $CaCO_3$ is facilitated as chromate anion ($CrO_4^{2-}$), which replaces carbonate anion ($CO_3^{2-}$) in the calcite lattice (Tang et al., 2007). This process of inorganic calcification tends to preferentially incorporate heavy $^{53}Cr$ isotopes into the mineral, yielding a $\delta^{53}Cr$ of calcite which is thus up to ∼0.3‰ more positive compared to the fluid (Rodler et al., 2015). In contrast, biologically produced marine $CaCO_3$ minerals, such as foraminiferal calcite (Wang et al., 2016), coral aragonite (Pereira et al., 2015) and bulk carbonate sediments (Holmden et al., 2016), are all systematically enriched in light Cr isotopes compared to ambient seawater. This fractionation trend is thus the exact opposite of the situation observed in inorganic calcite (cf., Rodler et al., 2015).

Furthermore, due to local redox cycling and biological uptake of Cr in the oceans (Semeniuk et al., 2016), the Cr isotope signature of present-day seawater is not globally homogeneous (Scheiderich et al., 2015; Paulukat et al., 2016), further complicating the application of the $\delta^{53}Cr$ proxy in marine carbonate archives. Considering the abovementioned issues and limitations, the full potential of Cr isotopes for paleo-redox studies can only be realized with more detailed calibration work done on the modern seawater-carbonate system from different oceanographic settings, by analyzing $\delta^{53}Cr$ data from both local seawater and precipitated marine carbonates.

This study is the first to present such a comprehensive Cr isotope investigation of the seawater-carbonate system from one of the world's largest carbonate-producing shelf ecosystems, the Great Barrier Reef (Lady Elliot Island, Australia), where $\delta^{53}Cr$ data were acquired from local ocean waters and selected recent biogenic carbonates (i.e., gastropods, cephalopods, corals, and calcifying algae). In addition, an alternative and complementary redox proxy, i.e., Cerium anomaly (Ce/Ce*), is applied here to further test and quantify the possible role of redox processes during Cr incorporation into marine biogenic carbonates. Elemental abundances of selected elements (Al, Mn, Fe, and REE), coupled with $^{87}Sr/^{86}Sr$ analysis, are also used as indices to evaluate possible contamination by non-marine Cr sources originating from lithogenic (detrital minerals, clays) and/or Mn–Fe oxide components (Pereira et al., 2015; Wang et al., 2016; Rodler et al., 2016; Gilleaudeau et al., 2016). The Cr isotope data acquired from the seawater-carbonate system at Lady Elliot Island are complemented by additional $\delta^{53}Cr$ analyses of marine skeletal carbonates (i.e., bivalves, gastropods) collected from the main oceanic provinces including: North and South Atlantic, North and South Pacific Oceans, and the Mediterranean Sea. Finally, these are then compared to published seawater $\delta^{53}Cr$ signatures for the above oceanic provinces (Scheiderich et al., 2015; Paulukat et al., 2016), and conclusions are made about the redox-controlled mechanism(s) behind the Cr isotope fractionation in a seawater-carbonate system, with implications for paleo-seawater $\delta^{53}Cr$ reconstructions.

**2. Study sites and samples**

Lady Elliot Island (24°06′47″S, 152°44′50″E) is the southernmost tropical coral cay of the Great Barrier Reef (GBR), which is the largest coral reef ecosystem on Earth with in-situ production of skeletal carbonates within an area comprising about 344,400 km², stretching along ∼2300 km off the coastline of Queensland, Australia (Fig. 1), (De'ath et al., 2012; Chivas et al., 1986). Lady Elliot Island (LEI) represents a particularly interesting and unique site for Cr isotope studies mostly because of (i) its remote location from the Australian mainland (ca. 80 km from the nearest coastline), and (ii) the effect of prevailing south-easterly trade winds blowing from the open ocean towards the mainland (Kench and Brander, 2006). Together these presumably result in a minimum input of possible continentally-derived Cr sources (i.e., mineral dust, silicate detritus) into the studied seawater-carbonate system. This unique setting of LEI, in turn, minimizes potential contamination of our samples with lithogenic/detrital Cr, which has been proven to impact $\delta^{53}Cr$ measurements in naturally Cr-poor marine carbonates (Pereira et al., 2015; Wang et al., 2016; Rodler et al., 2016).

The reef platform of LEI is kidney-shaped, measuring ∼1.2 km along its longest (NE-SW) and ∼0.8 km along its shorter (SE-NW) axis, with typical water depths ranging from <1 to about 25 m, across the reef-flat and reef-slope areas, respectively (Hamylton, 2014). Seawater samples analyzed in this study (Fig. 1C) were collected in 2015 by the LEI's Eco Resort scuba-diving team on (i) the leeward side of the island from the reef-flat and slope areas (near Coral Garden), and also on (ii) the windward side from the northeast lagoon (i.e., a reef-flat) and the northerly surf-break area (i.e., near a reef slope), (see also Table 1).

The main shallow-water calcifying organisms and benthic carbonate producers at the LEI's reef platform are: (i) scleractinian corals, (ii) red coralline algae, (iii) green calcifying algea (*Halimeda*), and marginally also (vi) benthic foraminifera, bivalves, and gastropods (Hamylton, 2014). Samples of recent biogenic carbonates were collected in 2013 from the leeward side of LEI, in an area called the 'Coral Gardens' (Fig. 1), and these specimens were sampled under the Queensland Museum general purposes permit.

**Table 1**

Ocean waters from Lady Elliot Island (LEI), the Great Barrier Reef, South Pacific Ocean (background data, chromium concentrations and isotope compositions).

| Sample ID | Material | Coordinates: latitude/longitude | Sampling site: other specifications | Depth (m) | Cr (ng/kg) | $\delta^{53}$Cr (‰) | 2sd | $n$ |
|---|---|---|---|---|---|---|---|---|
| SW 1 | Seawater | 24°06′35″S, 152°42′48″E | Leeward side/Reef flat | ∼0.5 | 193 | 0.75 | 0.03 | 1 |
| SW 2 | Seawater | 24°06′31″S, 152°42′42″E | Leeward side/Reef slope | ∼0.5 | 191 | 0.80 | 0.01 | 2 |
| SW 3 | Seawater | 24°06′30″S, 152°42′58″E | Windward side/Reef flat | ∼0.5 | 202 | 0.72 | 0.10 | 2 |
| SW 4 | Seawater | 24°06′19″S, 152°42′56″E | Windward side/Reef slope | ∼0.5 | 173 | 1.01 | 0.08 | 2 |

[Figure]

**Fig. 1.** (**A**) Map showing the study site, i.e., Lady Elliot Island (LEI), located in the southern Great Barrier Reef (GBR), Australia. (**B**) An aerial view of LEI with indicated sampling sites for ocean waters (yellow circles) and marine biogenic carbonates (orange polygon). (**C**) Schematic map of the island showing different zones within the LEI's reef system (i.e., flat, rim, slope), also including the sampling locations for seawaters and marine carbonates. (For interpretation of the colors in the figure(s), the reader is referred to the web version of this article.)

Importantly, the studied biogenic carbonates from LEI contain all the major representatives of key carbonate producing organisms including: (i) scleractinian corals, (ii) calcifying algae, (iii) and numerous species of gastropods (Table 2; and Fig. A1 and Table A1, Appendix).

To explore the range of $\delta^{53}$Cr values, and their relationships to REE patterns in marine skeletal carbonates on a global scale, we also assembled and analyzed different species of recent bivalves and gastropods collected from numerous locations worldwide, i.e. other locations (OL), which represent the main oceanic provinces including: North and South Atlantic, North and South Pacific Oceans, and the Mediterranean Sea (Table 3; Fig. A2 and Table A2, Appendix).

**3. Methods**

**3.1. Sample preparation**

Seawater samples from LEI were collected into pre-cleaned plastic bottles and stored in a fridge until elemental and isotope analyses. Prior to analysis the samples were filtered through 0.45 μm nylon membrane filters (Advantec MFS) and acidified.

Biogenic carbonates used for the isotope and elemental analysis were first physically scarped and washed in Milli-Q water (MQ, resistivity 18 Mohm/cm), and the shell fragments were then immersed in hydrogen peroxide ($H_2O_2$) and ethanol ($C_2H_6O$), followed by a brief leach in 1 N HCl and a final thorough washing in MQ water (following cleaning procedure P-5 from Zaky et al., 2015).

**3.2. Elemental concentration analysis (major, trace and REE)**

Major and trace element concentrations in biogenic carbonates were analyzed, respectively, by ICP-OES (Agilent 5100) and a sector field ICP-MS Element II (Thermo Fisher Scientific) at the Institute of Geology, Czech Academy of Sciences (for details see Appendix). The REE concentrations in filtered seawater and pre-cleaned LEI carbonates were determined using a seaFAST online pre-concentration system (Elemental Scientific Inc., USA) and ICP-MS (Agilent 7500ce) at GEOMAR. Our methods followed the analytical approach of Hathorne et al. (2012) and Osborne et al. (2017), for seawater and carbonate samples, respectively. Details on analytical errors, reproducibility, detection limits and procedural blanks of the seaFAST REE analyses are available in the Appendix (Tables A3 and A4).

The shale-normalized cerium anomaly (Ce/Ce*) reported in this study is calculated based on the following equation (Webb and Kamber, 2000, and Bau and Dulski, 1996):

Ce/Ce*

$$= [Ce/Ce_{(PAAS)}] / [0.5 \times (La/La_{(PAAS)}) + 0.5 \times (Pr/Pr_{(PAAS)})] \quad (1)$$

where PAAS represents the 'Post-Archean Average Australian Shale' (cf., Taylor and McClennan, 1985).

**3.3. Strontium isotope analysis ($^{87}Sr/^{86}Sr$) by TIMS**

Selected samples of biogenic carbonates and seawaters (both filtered and unfiltered) from LEI were analyzed for $^{87}Sr/^{86}Sr$ ratios by TIMS (VG Sector 54 IT) at the University of Copenhagen. Briefly, Sr was purified from samples using SrSpec resin, and eluted Sr was loaded on single Re filaments with a $Ta_2O_5$–$H_3PO_4$–HF activator. The presented $^{87}Sr/^{86}Sr$ values were corrected for the offset relative to the certified NIST SRM 987 of 0.710248 (McArthur et al., 2006). The reported errors ($2\sigma$) are within-run precisions of the individual analysis.

Please cite this article in press as: Farkaš, J., et al. Chromium isotope fractionation between modern seawater and biogenic carbonates from the Great Barrier Reef, Australia: Implications for the paleo-seawater $\delta^{53}$Cr reconstruction. Earth Planet. Sci. Lett. (2018), https://doi.org/10.1016/j.epsl.2018.06.032

**Table 2**

Marine biogenic carbonates from Lady Elliot Island (LEI), the Great Barrier Reef, South Pacific Ocean (background data, mineralogy, elemental and isotope compositions).

| Sample ID | Carbonate mineralogy | Coordinates: latitude/longitude | Organism order/phylum | Cr (ppm) | $\delta^{53}$Cr (‰) | 2sd | $n$ | Al (ppm) | Fe (ppm) | Mn (ppm) | $D_{Cr}$ (c/sw) |
|---|---|---|---|---|---|---|---|---|---|---|---|
| LEI 1 | Aragonite | 24°06′33″S 152°42′56″E | *Vetigastropoda* Mollusca | 0.042 | 0.31 | 0.08 | 2 | 5.0 | 11.2 | 1.1 | 221 |
| LEI 2 | Aragonite | 24°06′33″S 152°42′56″E | *Scleractinia* Cnidaria – Coral | 0.148 | 0.40 | 0.06 | 3 | 34.7 | 47.6 | 1.7 | 779 |
| LEI 3 | Aragonite | 24°06′33″S 152°42′56″E | *Scleractinia* Cnidaria – Coral | 0.096 | 0.32 | 0.06 | 7 | 15.3 | 55.8 | 0.9 | 505 |
| LEI 4 | Aragonite | 24°06′33″S 152°42′56″E | *Sepiida* Mollusca | 0.076 | 0.34 | 0.07 | 4 | 10.5 | 30.7 | 1.8 | 400 |
| LEI 5 | Aragonite | 24°06′33″S 152°42′56″E | *Caenogastrop.* Mollusca | 0.072 | 0.61 | 0.05 | 3 | 6.3 | 8.3 | 0.8 | 379 |
| LEI 6 | Aragonite | 24°06′33″S 152°42′56″E | *Vetigastropoda* Mollusca | 0.034 | 0.35 | 0.08 | 2 | 4.8 | 23.2 | 1.1 | 179 |
| LEI 7 | Aragonite + calcite (HMC) | 24°06′33″S 152°42′56″E | *Neritimorpha* Mollusca | 0.015 | 0.36 | 0.09 | 2 | 16.3 | 63.4 | 3.8 | 79 |
| LEI 8 | Aragonite | 24°06′33″S 152°42′56″E | *Caenogastrop.* Mollusca | 0.017 | 0.31 | 0.09 | 1 | 3.1 | 32.8 | 2.4 | 89 |
| LEI 9 | Aragonite | 24°06′33″S 152°42′56″E | *Vetigastropoda* Mollusca | 0.062 | 0.29 | 0.07 | 3 | 9.8 | 30.5 | 0.9 | 326 |
| LEI 10 | Calcite (HMC) | 24°06′33″S 152°42′56″E | *Coralline Algae* Rhodophyta | 0.450 | 0.86 | 0.07 | 6 | 10.9 | 40.7 | 3.8 | 2356 |
| LEI 11 | Calcite (HMC) | 24°06′33″S 152°42′56″E | *Alcyonacea* Cnidaria – Coral | 2.081 | 0.42 | 0.06 | 7 | 58.4 | 61.4 | 3.0 | 10895 |
| LEI 12 | Aragonite + calcite (HMC) | 24°06′33″S 152°42′56″E | *Coralline Algae* Rhodophyta | 0.191 | 0.45 | 0.07 | 6 | 10.5 | 21.1 | 1.6 | 1000 |

**3.4. Chromium isotope analysis ($\delta^{53/52}$Cr) by TIMS**

The acid-digested samples (i.e., filtered seawaters and pre-cleaned carbonates) were spiked with $^{50}$Cr–$^{54}$Cr tracer (i.e., double spike) and processed through a two-step Cr purification chromatography, using a combination of anionic and cationic exchange columns. Purified Cr fractions were analyzed on an IsotopX Phoenix thermal ionization mass spectrometer (TIMS) at the University of Copenhagen (Frei et al., 2009; Pereira et al., 2015). The Cr isotope composition of a sample is expressed as $\delta^{53}$Cr notation (normalized to SRM 979) in per mil (‰), according to the following equation:

$$\delta^{53}\text{Cr (‰)}$$
$$= \left[ \left(^{53}\text{Cr}/^{52}\text{Cr}\right)_{\text{SAMPLE}} / \left(^{53}\text{Cr}/^{52}\text{Cr}\right)_{\text{SRM979}} - 1 \right] \times 1000 \qquad (2)$$

The external reproducibility of the $\delta^{53}$Cr data is about $\pm 0.05$‰ (2sd), based on repeated measurements of certified standards (SRM 979, JDo-1 and JLs-1) and samples (Tables 2 and 3).

**4. Results**

**4.1. Oceanic waters – chromium isotope compositions ($\delta^{53}$Cr) and elemental concentrations**

A total of 4 seawater samples (i.e., South Pacific ocean waters) collected at different sites in the vicinity of Lady Elliot Island (Fig. 1C) were analyzed for $\delta^{53}$Cr, Cr and REE concentrations (see data in Table 1; and Table A5, Appendix). The Cr concentrations of LEI seawater range from 173 to 202 ng/kg, and their $\delta^{53}$Cr spread from ~0.72‰ to ~1.01 $\pm$ 0.05‰ (Fig. 2A), thus overlapping with published data for 'global surface ocean' waters collected from other locations (Paulukat et al., 2016; Scheiderich et al., 2015). Importantly, our results reveal a strong and statistically significant correlation ($r^2 = 0.96$, $p < 0.05$) between $\delta^{53}$Cr and concentration [Cr] data in LEI seawaters (Fig. 2B, C). Such coupling between Cr isotope compositions and concentrations has also been observed in ocean waters collected worldwide (Paulukat et al., 2016; Scheiderich et al., 2015). But the latter global dataset yields a specific slope of $-0.79$ which thus differs from that of the LEI's seawaters. The latter yield a slope of $-1.93 \pm 0.27$ ($1\sigma$) in a cross-plot with $\delta^{53}$Cr vs. ln[Cr] coordinates (Fig. 2C). The REE analysis of LEI seawaters (with the exception of a sample SW 1) show patterns typical for coastal seawater from the Great Barrier Reef, with characteristic enrichments of heavy REE and generally less negative Ce/Ce* anomalies ranging from ~0.322 to ~0.592 (i.e., data from this study and Wyndham et al., 2004), compared to typical seawater from the Pacific Ocean with a more pronounced negative Ce/Ce* anomaly of ~0.054 (see data in Tables A7 and A8, Appendix).

**4.2. Biogenic carbonates – chromium isotope variations ($\delta^{53}$Cr) and elemental concentrations**

A total of 29 recent marine biogenic carbonates were analyzed for $\delta^{53}$Cr values and elemental concentrations (i.e., Cr, Mg, Sr, Al, Fe, Mn, and REE), including 12 samples from LEI (Table 2), and 17 skeletal carbonates originating from 'other locations' (OL) worldwide. For details see also the Appendix and elemental data for LEI and OL samples listed in Tables A5 and A6, respectively (REE data are in Tables A9 and A10).

Overall, the investigated biogenic carbonates ($n = 29$) yielded Cr concentrations ranging from ~0.008 to ~2.081 ppm, and their $\delta^{53}$Cr values range from ~0.05‰ to ~0.86 $\pm$ 0.07‰, with no statistically significant correlation between $\delta^{53}$Cr and Cr concentration data ($r^2 = 0.29$, $p = 0.12$). Nevertheless, there is an apparent systematic difference between $\delta^{53}$Cr signatures of carbonates from LEI, and those from OL ($p < 0.0001$). The former yields an average $\delta^{53}$Cr of ~0.42 $\pm$ 0.16‰ (1sd, $n = 12$), whereas the latter sample set gives a much lower average of ~0.14 $\pm$ 0.12‰ (1sd, $n = 17$) (Tables 2 and 3). There seems to be no obvious correlation between $\delta^{53}$Cr signatures and Al or Fe concentrations (or Al/Cr and Fe/Cr elemental ratios) of the investigated carbonates

Please cite this article in press as: Farkaš, J., et al. Chromium isotope fractionation between modern seawater and biogenic carbonates from the Great Barrier Reef, Australia: Implications for the paleo-seawater $\delta^{53}$Cr reconstruction. Earth Planet. Sci. Lett. (2018), https://doi.org/10.1016/j.epsl.2018.06.032

**Table 3**

Marine biogenic carbonates from other locations (OL) worldwide, including Mediterranean Sea, North and South Atlantic, and North Pacific Ocean (background data, elemental and isotope compositions).

| Sample ID | Oceanic/sea water body | Location country | Coordinates: latitude/longitude | Organism class/phylum | Cr (ppm) | $\delta^{53}$Cr (‰) | 2sd | $n$ | Al (ppm) | Fe (ppm) | Mn (ppm) |
|---|---|---|---|---|---|---|---|---|---|---|---|
| OL 1 | Mediterranean | *Italy* Sicily | 37°59′43″N 13°40′30″E | *Bivalvia* Mollusca | 0.041 | 0.20 | 0.09 | 4 | 1.5 | 4.3 | 2.1 |
| OL 2 | Mediterranean | *Italy* Sicily | 37°59′43″N 13°40′30″E | *Gastropoda* Mollusca | 0.059 | 0.16 | 0.06 | 4 | 3.6 | 2.8 | 3.7 |
| OL 3 | Mediterranean | *Italy* Bellaria | 44°07′56″N 12°29′16″E | *Gastropoda* Mollusca | 0.180 | 0.08 | 0.06 | 9 | 8.6 | 9.5 | 27.6 |
| OL 4 | Mediterranean | *Italy* Bellaria | 44°07′56″N 12°29′16″E | *Bivalvia* Mollusca | 0.510 | 0.05 | 0.10 | 9 | 15.4 | 20.7 | 35.0 |
| OL 5 | Mediterranean | *Italy* Bellaria | 44°07′56″N 12°29′16″E | *Bivalvia* Mollusca | 0.047 | 0.10 | 0.08 | 7 | 1.8 | 12.1 | 10.7 |
| OL 6 | North Atlantic | *Brittany* Carnac | 47°34′14″N 03°04′31″W | *Gastropoda* Mollusca | 0.098 | 0.19 | 0.07 | 6 | 2.1 | 28.0 | 6.8 |
| OL 7 | North Atlantic | *Brittany* Carnac | 47°34′14″N 03°04′31″W | *Gastropoda* Mollusca | 0.008 | 0.06 | 0.20 | 1 | 1.3 | 0.9 | 1.5 |
| OL 8 | North Atlantic | *Brittany* Brest | 48°17′33″N 04°27′14″W | *Gastropoda* Mollusca | 0.085 | 0.12 | 0.06 | 7 | 4.6 | 53.4 | 0.7 |
| OL 9 | North Atlantic | *Brittany* Carnac | 47°34′14″N 03°04′31″W | *Bivalvia* Mollusca | 0.015 | 0.08 | 0.09 | 1 | 1.7 | 22.5 | 1.9 |
| OL 10 | North Atlantic | *Norway* Oslo | 59°19′36″N 10°40′06″E | *Bivalvia* Mollusca | 0.053 | −0.02 | 0.06 | 3 | 3.1 | 35.1 | 4.0 |
| OL 11 | South Atlantic | *Argentina* Viedma | 41°09′20″S 63°07′49″W | *Gastropoda* Mollusca | 0.064 | 0.20 | 0.08 | 4 | 33.1 | 152.4 | 3.1 |
| OL 12 | South Atlantic | *Argentina* Viedma | 41°09′20″S 63°07′49″W | *Bivalvia* Mollusca | 0.025 | 0.14 | 0.08 | 1 | 1.9 | 3.1 | 23.5 |
| OL 13 | South Atlantic | *Argentina* Viedma | 41°09′20″S 63°07′49″W | *Gastropoda* Mollusca | 0.043 | 0.25 | 0.11 | 3 | 3.6 | 3.6 | 1.6 |
| OL 14 | North Pacific | *USA* Alaska | 56°50′37″N 134°00′24″W | *Gastropoda* Mollusca | 0.017 | 0.10 | 0.09 | 2 | 0.6 | 3.4 | 0.6 |
| OL 15 | North Pacific | *USA* Alaska | 56°50′37″N 134°00′24″W | *Bivalvia* Mollusca | 0.272 | 0.54 | 0.07 | 12 | 0.3 | 23.9 | 0.1 |
| OL 16 | North Pacific | *USA* Alaska | 56°50′37″N 134°00′24″W | *Bivalvia* Mollusca | 0.048 | 0.11 | 0.06 | 6 | 2.3 | 49.0 | 0.5 |
| OL 17 | North Pacific | *USA* Oregon | 44°29′11″N 124°05′05″W | *Gastropoda* Mollusca | 0.130 | 0.07 | 0.07 | 7 | 15.9 | 15.1 | 3.3 |

[Figure]

**Fig. 2.** (**A**) The Cr isotope compositions ($\delta^{53}$Cr) and concentrations (ng/kg) in ocean waters collected from Lady Elliot Island (LEI), with a linear regression trend (i.e., red line) fitted to seawater data. (**B**) $\delta^{53}$Cr versus 1/[Cr] with logarithmic trends through data, where the red line represents a log fit function for LEI seawater samples, and the blue trend was constructed using the 'global seawater dataset' compiled by Paulukat et al. (2016); see also references therein. (**C**) $\delta^{53}$Cr versus ln[Cr] cross-plot with linear regression trends: red line = LEI data; and blue line = Global Ocean Water dataset from Paulukat et al. (2016).

from LEI and OL (Fig. A4), which might suggest a lack of contamination from detrital clays and/or Fe-oxides (Wang et al., 2016; Pereira et al., 2015). However, a cross-plot of Cr isotopes versus Mn/Cr ratios (Fig. A5) reveals that biogenic carbonates from OL yield systematically lower $\delta^{53}$Cr and relatively higher Mn/Cr, compared to samples from LEI. This difference was also confirmed by the non-parametric Mann–Whitney U statistic test (an equivalent for the parametric Student t-test; for details see the Appendix). This, in turn, points to a possible contamination issue related to Mn-oxide coatings on the measured Cr isotope compositions of carbonates collected from other locations (OL), which are indeed expected to be more impacted by terrestrial inputs and lithogenic Cr sources (due to their proximity to continents) relative to samples from LEI.

As for REE, the samples from LEI exhibit rather low $\Sigma$REE ranging from ∼1.5 to 93 (ppb), and variable true negative Ce/Ce* (PAAS) anomalies from ∼0.298 up to ∼0.825 (Table A9), whose fidelity was confirmed also via Ce/Ce* vs. Pr/Pr* and La/Ce cross-plots (see Fig. A3, Appendix). A very similar range was observed in the samples from OL, which yielded Ce/Ce* from ∼0.375 to ∼0.925, but

Please cite this article in press as: Farkaš, J., et al. Chromium isotope fractionation between modern seawater and biogenic carbonates from the Great Barrier Reef, Australia: Implications for the paleo-seawater $\delta^{53}$Cr reconstruction. Earth Planet. Sci. Lett. (2018), https://doi.org/10.1016/j.epsl.2018.06.032

**Fig. 3.** (**A**) Ce/Ce* and $\Sigma$REE for pre-cleaned LEI and OL biogenic carbonates (this study), and datasets from different cleaning procedures performed on modern biogenic carbonates (i.e., brachiopod shells) by Zaky et al. (2015). The latter study includes 'thoroughly cleaned' samples (i.e., P4 and P5 datasets) that involved a combination of physical and chemical cleaning (i.e., scraping, sonication in Milli-Q water and $H_2O_2$, and leaching in HCl acid); and 'partially cleaned' samples that only went through physical cleaning and/or $H_2O/H_2O_2$ washing (i.e., P1, P2 and P3 datasets; for details see Zaky et al., 2015). (**B**) A close-up view of the datasets presented in Fig. 3A (see the dashed rectangle). (**C**) A cross-plot of Mn vs. Fe concentrations (ppm) in LEI and OL biogenic carbonates, and datasets from different cleaning procedures of Zaky et al. (2015).

their $\Sigma$REE are systematically higher, i.e., ranging from ∼37 up to 998 (ppb), (Table A10). Interestingly, the biogenic carbonates from LEI (i.e., benthic molluscs) reveal a strong and statistically significant negative correlation ($r^2 = 0.83$, $p < 0.05$, $n = 6$) between $\delta^{53}$Cr and Ce/Ce*, the latter being a redox proxy. A very similar relationship between Cr isotopes and Ce/Ce* anomaly data was observed recently by Bonnand et al. (2013) in modern marine ooids from the Bahamas ($r^2 = 0.93$, $p < 0.05$, $n = 4$). However, no such correlation between $\delta^{53}$Cr and Ce/Ce* was observed in our samples of corals and calcareous algae from LEI, nor in our data from the OL samples (see Appendix, Fig. A6).

**4.3. Strontium isotopes ($^{87}Sr/^{86}Sr$) in seawaters and biogenic carbonates from LEI**

Seawater collected from LEI (filtered and unfiltered) yielded $^{87}$Sr/$^{86}$Sr ratios ranging from 0.709162 to 0.709171, with a mean of 0.709166 ± 0.000005 ($2\sigma$, $n = 10$; Table A12, Appendix), which is consistent with present-day global ocean water (McArthur et al., 2006; Pereira et al., 2015). As for biogenic carbonates from LEI (Table A13), their $^{87}$Sr/$^{86}$Sr ranges from 0.709161 to 0.709175, giving a mean of 0.709168 ± 0.000008 ($2\sigma$, $n = 24$), which thus overlaps within error with the mean $^{87}$Sr/$^{86}$Sr of LEI seawater (Fig. 4).

**5. Discussion**

**5.1. Evaluating the effect of contamination by detrital/lithogenic components and Mn/Fe oxide coatings via $\Sigma$REE, trace elements and $^{87}Sr/^{86}Sr$ indices**

Numerous studies have shown that adsorbed detritus and/or oxide coatings on marine biogenic carbonates may lead to (i) elevated $\Sigma$REE concentrations, (ii) positive and non-marine Ce/Ce* values, and/or (iii) increased Mn and Fe concentrations (Zaky et al., 2015, and references therein). To evaluate these effects and their possible impact on the measured Ce/Ce* and Cr isotope variations in the studied biogenic carbonates, we adopted the approach of Zaky et al. (2015) and plot our data in Ce/Ce* vs. $\Sigma$REE, and Mn vs. Fe concentration plots (Fig. 3).

For the purposes of comparison, we present our results (i.e., LEI and OL samples) along with data from Zaky et al. (2015) that represent (i) 'thoroughly cleaned' carbonate shells (i.e., physical and chemical treatment = P4 and P5 procedures), and (ii) 'partially cleaned' samples, where the detrital components and oxide coatings were not properly removed (i.e., P1, P2 and P3 procedures).

Importantly, this comparison reveals that our data from LEI and OL carbonates do not show any obvious correlation between Ce/Ce* and $\Sigma$REE, where the latter would be indicative of contamination issues. Overall our data plot within the range of $\Sigma$REE, Mn and Fe concentrations measured in the 'thoroughly cleaned' samples of Zaky et al. (2015). This in turn points to the effectiveness and suitability of our selected cleaning procedure (i.e. P5, as defined by Zaky et al., 2015).

However, the OL samples exhibit generally higher $\Sigma$REE and Mn/Cr ratios, coupled with systematically lower $\delta^{53}$Cr values (compared to the LEI samples; see Fig. 3B, Fig. A5), and these differences between OL and LEI data sets are statistically significant, as confirmed by the Mann–Whitney test (see the Appendix). Such elevated Mn/Cr coupled with low $\delta^{53}$Cr values, the latter approaching −0.1 per mil (i.e., the Cr isotope signature of the Earth's silicate crust; Schoenberg et al., 2008; Farkaš et al., 2013), might thus indicate possible contamination of OL samples by lithogenic Cr sources, possibly related to the presence of Mn-oxide coatings. We speculate that perhaps Mn oxyhydroxides could impact the $\delta^{53}$Cr of marine skeletal carbonates due to oxidation of lithogenic and isotopically light Cr(III) present in local sediments and pore waters to more soluble and mobile Cr(VI) species, and their subsequent incorporation into carbonate shells. The latter could be facilitated either via (i) adsorption of such sediment-derived Cr onto reactive surfaces of oxide coatings or directly onto a shell, or alternatively via (ii) chemical exchange and diffusion of Cr between the shell and local pore fluids at the sediment–seawater interface. These hypothesized pathways can thus explain the generally 'non-marine' and very low Cr isotope signatures observed in the OL samples and their association with elevated Mn concentrations (Fig. A5, Table A6). Note that pore waters are expected to have $\delta^{53}$Cr signatures close to ∼0‰ due to a large reservoir of lithogenic Cr hosted in marine sediments that are typically dominated by siliciclastic detrital and/or clay minerals.

Thus, our cleaning procedure seems to be sufficient for the LEI carbonates, which originated from a pristine and remote marine setting with minimum input of detrital and lithogenic Cr sources. This procedure seems, however, to be inadequate for a thorough cleaning and removal of lithogenic Cr from the OL carbonate samples, likely because these have been more exposed and impacted by non-marine Cr sources. Overall, our data seem to indicate that a large part of Cr hosted in the shells from OL might have originated from lithogenic or sediment/pore water derived Cr sources,

[Figure]

**Fig. 4.** Strontium isotope ratios ($^{87}$Sr/$^{86}$Sr) measured in biogenic carbonates (LEI data) and seawater (SW) collected from Lady Elliot Island. For details see also data listed in Tables A12 and A13. The following abbreviations are used: SW = seawater, F = filtered, UF = unfiltered. Note that for carbonates (i.e., LEI data), two aliquots (a, b) from the same sample were processed via columns and analyzed for $^{87}$Sr/$^{86}$Sr.

which thus overprinted the primary and isotopically heavy marine $\delta^{53}$Cr signal in certain shells.

The above diagenetic interpretation is supported by a single sample of bivalve shell from the OL dataset (i.e., a sample OL 15, from Alaska, North Pacific) that yielded the heaviest Cr isotope composition of +0.54‰, which is thus very close to the expected $\delta^{53}$Cr of ambient seawater (Paulukat et al., 2016). This sample also has the lowest Mn and Al concentrations of all of the OL carbonates analyzed (Tables A6, Figs. A4 and A5). Hence, it seems that apart from this sample, which records a close-to original seawater $\delta^{53}$C signal, the rest of OL shells seem to have their Cr isotope compositions impacted to a variable degree by secondary processes, involving later incorporation of lithogenic Cr mediated by the presence of Mn in the system.

Consequently, our further discussion of Cr isotope variations between seawater and biogenic carbonates is focused primarily on the samples from LEI, which showed minimum or no obvious impact of lithogenic and/or Mn–Fe oxide coating contamination on their measured $\delta^{53}$Cr and Ce/Ce* values. To further corroborate the lack of secondary contamination in the LEI samples, we adopted the approach of Pereira et al. (2015) and also analyzed $^{87}$Sr/$^{86}$Sr in biogenic carbonates and local seawater from the Lady Elliot Island (Tables A12, A13). The above isotope proxy is useful, as shifts to more radiogenic $^{87}$Sr/$^{86}$Sr values in marine carbonates (relative to seawater) would indicate the presence of lithogenic/detrital contamination. Importantly, our data confirmed no detectable radiogenic or non-marine Sr sources in the studied LEI carbonates. As the latter yielded $^{87}$Sr/$^{86}$Sr ratios that are identical (within the analytical error) to the local seawater (see Fig. 4), which in turn is consistent with the global ocean Sr isotope composition. Considering the above constraints from Sr isotopes, as well as Ce/Ce* vs. ΣREE, and Mn vs. Fe concentration plots (Fig. 3, and Fig. A5), we are confident that the geochemical data acquired from the LEI carbonates have not been affected by contamination and early diagenetic processes. This is likely related to the fact that LEI samples originate from a remote carbonate-dominated and detrital-poor marine setting of the Great Barrier Reef, while OL samples are from generally carbonate-poor coastal areas more affected by continental inputs (see Table 3). Therefore, we conclude that $\delta^{53}$Cr and Ce/Ce* values measured in LEI carbonates reflect the primary signal related to redox cycling of Cr and Ce in a local marine environment, and/or micro-environment associated with the biological uptake of these redox-sensitive elements during biocalcification.

**5.2. Chromium isotope composition of LEI seawater and the impact of a local redox cycling?**

Recent studies have revealed significant heterogeneities with respect to Cr cycling in the oceans, where $\delta^{53}$Cr of seawater ranges from ∼0.2‰ up to ∼1.6‰ as a function of water depth (Scheiderich et al., 2015) and/or geographical location (Paulukat et al., 2016). Importantly, these changes in $\delta^{53}$Cr are tightly coupled to changes in seawater Cr concentrations, following a globally recognized correlation trend (i.e., a linear relationship) with a slope of −0.79 ± 0.06 (Paulukat et al., 2016) when plotted in $\delta^{53}$Cr versus ln[Cr] space (Fig. 2C). Assuming a closed-system Rayleigh fractionation of Cr isotopes in the oceans, the above slope of −0.79 could be interpreted as a common 'global' fractionation factor ($\varepsilon$) associated with the redox cycling of Cr in seawater, specifically with the reduction of Cr(VI) to Cr(III) in the surface oceans and/or oxygen minimum zones (Paulukat et al., 2016; Scheiderich et al., 2015). Accordingly, the fractionation factor ($\varepsilon$) for a closed-system behavior of Cr isotopes can be calculated based on the following relationship (cf., Scheiderich et al., 2015):

$$\Delta^{53}Cr_{SW} = \varepsilon \ln[Cr] \qquad (3)$$

where $\varepsilon = 1000 * (\alpha - 1)$ is the fractionation factor in ‰ associated with the reduction of Cr(VI) to Cr(III), and $\alpha$ is the fractionation coefficient; and $\Delta^{53}Cr_{SW}$ is the relative change (i.e., difference) in the $\delta^{53}$Cr signature of dissolved Cr remaining in the seawater. Thus, $\Delta^{53}Cr_{SW}$ versus ln[Cr] should have a slope of $\varepsilon$ (Scheiderich et al., 2015).

Consequently, the fractionation factor ($\varepsilon$) associated with the reduction of Cr(VI) to Cr(III) in marine settings can be determined from the logarithmic relationship between Cr concentrations (ln[Cr]) and isotope compositions ($\delta^{53}$Cr) in a suite of seawater samples (Fig. 2C), (Scheiderich et al., 2015; Smith and Kroopnick, 1981).

Interestingly, seawater from LEI also shows a strong coupling between $\delta^{53}$Cr and ln[Cr] data, but with a much steeper slope of −1.93 ± 0.27 (1σ) (Fig. 2C), suggesting significantly larger local isotope fractionation effects during the reduction of Cr(VI) to Cr(III). Alternatively, there could be a locally enhanced input of reoxidized Cr(VI), originating from the oxidation of isotopically light Cr(III) present in bottom pore waters and/or organically-complexed Cr sources (Paulukat et al., 2016; Semeniuk et al., 2016). With our

Please cite this article in press as: Farkaš, J., et al. Chromium isotope fractionation between modern seawater and biogenic carbonates from the Great Barrier Reef, Australia: Implications for the paleo-seawater $\delta^{53}$Cr reconstruction. Earth Planet. Sci. Lett. (2018), https://doi.org/10.1016/j.epsl.2018.06.032

[Figure]

**Fig. 5.** (**A**) Cr isotope variations ($\delta^{53}$Cr) versus Ce-anomaly data (Ce/Ce*) measured in recent biogenic carbonates (i.e., diamonds) and local seawater (circles) collected from Lady Elliot Island. The following abbreviations are used: LEI = Lady Elliot Island; SW = Seawater; GBR = Great Barrier Reef. The blue vertical rectangle illustrates the range of $\delta^{53}$Cr and Ce/Ce* values measured in LEI seawater (i.e., samples SW 2, 3 and 4); and the light and dark blue horizontal lines illustrates the range of Ce/Ce* anomalies measured in coastal seawaters from the GBR (i.e., data from this study and Wyndham et al., 2004). The grey arrows indicate the directions of negative versus positive Ce/Ce* anomalies, respectively, for Pacific seawater and non-marine (i.e., lithogenic/detrital) continental inputs (see data in Tables A7 and A8, Appendix). (**B**) $\delta^{53}$Cr versus Ce/Ce* data from LEI molluscan shells (i.e., gastropods = green diamonds), and modern ooids from the Great Bahama Bank (i.e., red squares = data from Bonnand et al., 2013). The dashed green and red lines represent linear regression fits constructed for the LEI and Bahama datasets, respectively. Both show a statistically significant ($p < 0.05$) negative correlation between $\delta^{53}$Cr versus Ce/Ce* data. Black diamond represents an intercept value ($\delta^{53}$Cr ≈ +0.83‰) for the LEI's regression trend, which overlaps with the Cr isotope composition of LEI seawater (i.e., blue rectangle) with an average $\delta^{53}$Cr of +0.82 ± 0.13‰. A black square illustrates an intercept for Bahamas ooids data (Bonnand et al., 2013), which overlaps with the $\delta^{53}$Cr of Caribbean seawater (i.e., purple rectangle) with an average of +1.13‰ (Holmden et al., 2016). Note that the presented correlation coefficients ($r^2$) and p-values are calculated for standard parametric tests (i.e., Pearson correlation), but even if evaluated by non-parametric tests (i.e. Kendall's tau correlation analysis) the p-value for LEI data is still ~0.05, thus confirming the statistical significance of the observed $\delta^{53}$Cr versus Ce/Ce* correlation trend.

limited dataset, we cannot differentiate between the above scenarios (i.e., a local mixing versus Rayleigh). Future detailed studies of the Cr isotope cycling in shallow-water reef environments could help to better understand these local redox processes, their controlling factors, and effects on the Cr isotope composition of seawater (for details see also the discussion in Appendix).

**5.3. Evidence for redox-controlled incorporation of Cr into marine biogenic carbonates**

**5.3.1. Lady Elliot Island – chromium isotopes ($\delta^{53}$Cr) versus Cerium anomalies (Ce/Ce*)**

To further investigate a possible role of biologically mediated redox processes during the incorporation of Cr from seawater and/or a calcifying fluid into biogenic carbonate, we investigated the relationship between $\delta^{53}$Cr and Ce/Ce* data in our samples, as the latter proxy represents a sensitive redox indicator in marine settings (de Baar et al., 1985; Zaky et al., 2015). Briefly, under reducing marine conditions (i.e., anoxic and suboxic settings), Ce in seawater will be present mostly as soluble Ce(III) species, but in oxidizing environments the removal of Ce via uptake by Mn-oxides and its subsequent precipitation as insoluble Ce(IV) will be promoted (Elderfield, 1988; German and Elderfield, 1990). Due to the above redox-controlled partitioning of Ce in seawater, the oxidative removal of Ce from a solution can be quantified via measured Ce abundances normalized to the rest of REE concentration data (i.e., Ce/Ce* anomaly, see Eq. (1); Bau and Dulski, 1996; Webb and Kamber, 2000). In addition, biological processes such as the microbial oxidation of Ce in marine environments play also an important role in controlling the local redox cycling of Ce in seawater, and/or calcification fluids from which marine carbonates precipitate (Moffett, 1990; Wyndham et al., 2004).

Interestingly, our results from LEI biogenic carbonates, specifically benthic molluscan shells (i.e., gastropods), reveal a strong and statistically significant negative correlation between Ce/Ce* and $\delta^{53}$Cr data ($r^2 = 0.83$, $p = 0.01$, $n = 6$; see Fig. 5A, B). In particular, shells with heavier Cr isotope compositions (approaching the $\delta^{53}$Cr of local seawater) yield more negative Ce anomalies (Ce/Ce* ~ 0.3) which in turn suggest relatively more oxic conditions, or oxidizing

micro-environment, at the site of calcification during the biological uptake of Ce and Cr from seawater (Fig. 5A). Alternatively, the observed coupling between Ce/Ce* and $\delta^{53}$Cr data in shells could also perhaps be related to specific biological activity and ontogenesis of an organism, causing a preferential uptake of Ce and light Cr isotopes into organic matter (or the expulsion from a calcifying fluid), leaving the composition of a shell with lower Ce/Ce* and higher $\delta^{53}$Cr.

It should be pointed out, however, that a very similar negative correlation between Ce/Ce* and $\delta^{53}$Cr data ($r^2 = 0.94$, $p = 0.03$, $n = 4$), with a similar slope, was also documented recently for modern non-skeletal (microbial) marine carbonates, i.e., ooids, from the Bahama Bank (Fig. 5B; data from Bonnand et al., 2013). Importantly, for both of these relationships (i.e., our shells and published ooids) the intercepts of Ce/Ce* versus $\delta^{53}$Cr correlation overlap with the measured Cr isotope composition of local ocean waters (Fig. 5B). Specifically, the intercept for LEI shells is +0.83 ± 0.10 ($1\sigma$), and local LEI seawater yielded an average $\delta^{53}$Cr of +0.82 ± 0.13‰ ($\sigma$, $n = 4$), (see data in Table 1). As for Bahamas samples, the intercept for ooids is +1.09, while the published data for Caribbean seawaters have an average $\delta^{53}$Cr of +1.14 ± 0.03‰ ($\sigma$, $n = 3$), (Holmden et al., 2016).

These purported agreements between $\delta^{53}$Cr of local seawaters and intercepts derived from Ce/Ce* vs. $\delta^{53}$Cr relationships (Fig. 5) perhaps suggest that marine biogenic carbonates that precipitated under more oxidizing conditions or micro-environments (i.e., yielding the most negative Ce anomaly) might actually reflect the Cr isotope signature, approaching the composition of ambient ocean water. The latter might then be inferred from the intercept of Ce/Ce* vs. $\delta^{53}$Cr correlation acquired from a set of suitable marine carbonate archives (e.g., molluscan shells and/or microbial ooids). If validated by future studies, this approach could be used to reconstruct the $\delta^{53}$Cr signature of paleo-seawater based on Ce/Ce* and $\delta^{53}$Cr data in a set of well-preserved fossil marine carbonates collected at a specific site.

Interestingly, the only calcifying organism from the LEI location that yielded identical $\delta^{53}$Cr and Ce/Ce* values to those measured in ambient ocean water was a microbial carbonate (i.e., high-

Please cite this article in press as: Farkaš, J., et al. Chromium isotope fractionation between modern seawater and biogenic carbonates from the Great Barrier Reef, Australia: Implications for the paleo-seawater $\delta^{53}$Cr reconstruction. Earth Planet. Sci. Lett. (2018), https://doi.org/10.1016/j.epsl.2018.06.032

**Fig. 6.** Conceptual models for the isotope fractionation and redox cycling of Cr during biological uptake and calcification by marine organisms. (**A**) *Rayleigh model* with a finite reservoir of unreacted Cr(VI) pool in calcifying fluids, which does not interact with the Cr(III) product. (**B**) *Equilibrium* (i.e., steady-state) model with a constant exchange between the unreacted Cr(VI) and produced Cr(III). *Solid blue lines/curves* = the evolving $\delta^{53}$Cr signature of the remaining (i.e., unreacted) Cr(VI) reservoir in the system. *Dashed blue lines/curves* = the $\delta^{53}$Cr signature of an instantaneous Cr(III) product, from the reduction of Cr(VI) to Cr(III). *Redox Trajectories:* 1–2 = Partial reduction of Cr(VI) to Cr(III); 2–3 = Production of Cr(III) with an associated isotope fractionation ($\varepsilon$) of $-0.80$‰ for the reduction of Cr in marine settings (cf., Paulukat et al., 2016; Scheiderich et al., 2015). 3–4 = Incorporation of chromium (either as Cr(III), or as re-oxidized Cr(VI) species) from calcifying fluids into CaCO$_3$ biominerals. The initial Cr isotope composition used in our calculations (for local LEI seawater) has the $\delta^{53}$Cr value ≈ 0.82‰ (i.e., a solid black circle labeled '1'). Green crosses (X) represent the $\delta^{53}$Cr values measured in molluscan shells and microbial carbonates collected from LEI (see also data in Table 2), a black cross illustrates a specific molluscan sample LEI 05 with $\delta^{53}$Cr of ~0.61‰, that is modeled here considering the above *redox trajectories*. Note that a red cross with the highest $\delta^{53}$Cr represents a sample of microbial carbonate produced by red coralline algae (LEI 10) that yielded $\delta^{53}$Cr and Ce/Ce* which are identical with values found in local LEI seawater, suggesting a direct incorporation of Cr(VI) from seawater into CaCO$_3$ without additional biological/redox isotope fractionation.

Mg calcite) produced by calcifying red coralline algae (*Lithothamnion* sp.). This organism thus seems to incorporate Cr isotopes and REE from seawater without additional biological discrimination and/or redox-controlled isotope fractionation during biocalcification. Hence, microbial carbonates produced by certain species of calcifying algae, and their fossil counterparts (i.e., algal microbialites), could perhaps also be used as unique natural archives to trace the paleo-seawater $\delta^{53}$Cr signature over geological time.

**5.4. Plausible mechanism(s) for Cr isotope fractionation in biogenic carbonates**

Pioneering studies by Wang et al. (2016) and Pereira et al. (2015) that investigated the Cr isotope fractionation between recent marine biogenic carbonates (e.g., corals, foraminifera, algae) and ambient seawater revealed that the above organisms preferentially incorporate lighter Cr isotopes into their CaCO$_3$ skeletons. These studies proposed that such enrichments in light Cr isotopes in present-day biogenic carbonates could be due to (i) the incorporation of organically complexed and particle-reactive Cr(III) species directly from seawater into CaCO$_3$, and/or (ii) via biologically-mediated redox cycling involving an initial *reduction* of marine Cr(VI) to isotopically light Cr(III), and its subsequent *re-oxidation* and incorporation into a shell. The following section models the latter scenario quantitatively, i.e., the *reduction/re-oxidation* pathway, and discusses plausible mechanisms for the observed coupling between redox-sensitive $\delta^{53}$Cr and Ce/Ce* tracers documented in the studied molluscan shells from LEI (Fig. 5B).

**5.4.1. Rayleigh Cr-isotope fractionation model for redox-controlled biocalcification**

Assuming that Cr is incorporated into CaCO$_3$ minerals primarily as oxidized Cr(VI) ions (i.e., chromate CrO$_4^{2-}$ oxyanions replacing carbonate CO$_3^{2-}$ anions; Tang et al., 2007; Pereira et al., 2015), then the entire variability of a purported redox-controlled correlation between $\delta^{53}$Cr and Ce/Ce* data in LEI molluscan shells (and/or Bahamas microbial ooids, Fig. 5B) could be explained via a partial reduction of marine Cr(VI) to Cr(III) by the organism, followed by biologically mediated re-oxidation of isotopically light Cr(III) to Cr(VI), and its eventual incorporation into CaCO$_3$ biominerals (cf., Pereira et al., 2015). The latter redox pathway (i.e.,

reduction and re-oxidation), and its effects on the $\delta^{53}$Cr proxy, can be described quantitatively via a Rayleigh fractionation model that simulates a progressive reduction of marine Cr(VI) to Cr(III) and the associated isotope fractionation effects. Here we consider two scenarios, i.e., a *Rayleigh* and *equilibrium* fractionation models (see below, Eqs. (5) and (6)), and our calculations are based on the Cr isotope composition of a local LEI seawater ($\delta^{53}$Cr ≈ 0.82‰) and a common fractionation factor ($\varepsilon$) of $-0.80$‰ for the reduction of Cr in marine settings (Paulukat et al., 2016; Scheiderich et al., 2015). Using this model parameterization, the associated changes in $\delta^{53}$Cr of a calcifying fluid, and by inference of the precipitated biogenic carbonates (i.e., molluscan shells), can be calculated.

For the closed system *Rayleigh* fractionation (Fig. 6A) with a finite reservoir of Cr(VI) in the calcifying pool, and no exchange between the reactant Cr(VI) and produced Cr(III) species, the evolving $\delta^{53}$Cr of a calcifying fluid (due to biologically-mediated Cr reduction) is modeled using the following equation (Ellis et al., 2002):

$$\delta^{53}Cr_{(VI)} = \left[ (\delta^{53}Cr_{INI} + 10^3) f^{(\alpha-1)} \right] - 10^3 \qquad (4)$$

where $\delta^{53}Cr_{INI}$ refers to the *initial* isotope composition of an unreacted Cr(VI) pool in seawater or a calcifying fluid; $f$ is the fraction of Cr(VI) remaining in the pool; and $\alpha$ is the fractionation coefficient for the biologically-mediated reduction of Cr(VI) to Cr(III), where the latter relates to the fractionation factor ($\varepsilon$) according to $\varepsilon \approx 1000 * \ln \alpha$.

**5.4.2. Equilibrium Cr-isotope fractionation model for redox-controlled biocalcification**

An alternative *equilibrium* model considers steady-state conditions where Cr(VI) and Cr(III) species interact (Fig. 6B), and here the $\delta^{53}$Cr of a remaining Cr(VI) reactant in calcifying fluid is calculated based on (Frings et al., 2016):

$$\delta^{53}Cr_{(VI)} = \varepsilon * (1 - f) \qquad (5)$$

and an instantaneous reduced Cr(III) product being:

$$\delta^{53}Cr_{(III)} \delta^{53}Cr_{INI} + \varepsilon * f \qquad (6)$$

JID:EPSL   AID:15135 /SCO

[Figure]

**Fig. 7.** A compilation of $\delta^{53}$Cr values from recent marine biogenic carbonates (color-coded symbols) and ambient seawater (vertical colored rectangles), collected from the following oceanic provinces: South Pacific (this study), North Pacific, South and North Atlantic, Caribbean and Mediterranean Seas (seawater data from Paulukat et al., 2016; Scheiderich et al., 2015; Holmden et al., 2016; and $\delta^{53}$Cr data from biogenic carbonates are from this study (see also Table 3), and from Bonnand et al., 2013).

In the context of the proposed *reduction* and *re-oxidation* pathway for Cr incorporation into biogenic carbonates, the above equations (Eqs. (5) and (6)) describe the first step involving the biologically-mediated *partial reduction* of marine Cr(VI) to Cr(III) at calcification sites (see also 'redox trajectories' 1–2 and 2–3 in Fig. 6). The second step (i.e., a trajectory 3–4) will then involve either (i) a direct incorporation of the produced and isotopically light Cr(III) into carbonate, or (ii) a quantitative and near complete *re-oxidation* of such light Cr(III) to Cr(VI) and its subsequent incorporation as isotopically light Cr(VI)-oxyanions into CaCO$_3$ (cf., Pereira et al., 2015; Wang et al., 2016). The latter scenario, i.e., the *re-oxidation* step (via Mn(IV) and/or microbially mediated) is however more likely, considering that under alkaline and high pH conditions typical for calcification sites, the more stable and dominant form of chromium in a solution is Cr(VI) (Pereira et al., 2015). Hence, as illustrated by 'redox trajectories' in Fig. 6, one can explain the entire variability in $\delta^{53}$Cr and Ce/Ce* data of LEI molluscan shells via the above two-step *reduction/re-oxidation* pathway.

**5.5. Chromium isotope fractionation between biogenic carbonate and seawater – a 'global ocean' perspective**

To evaluate the systematics of Cr isotope fractionation between marine biogenic carbonates and ocean water on a global scale, we present a compilation of $\delta^{53}$Cr datasets that include (i) our data from LEI and (ii) recent bivalves and gastropods from numerous locations worldwide (i.e., other locations (OL), Tables 2 and 3), which are complemented by (iii) published $\delta^{53}$Cr data from seawaters representing the main oceanic water bodies/provinces including: North and South Atlantic, North and South Pacific Oceans, Caribbean and Mediterranean Seas (Paulukat et al., 2016; Scheiderich et al., 2015; Holmden et al., 2016). As shown in Fig. 7, marine biogenic carbonates tend to be systematically enriched in light Cr isotopes relative to local ocean waters. However, some of these shifts to lower $\delta^{53}$Cr (i.e., those observed in OL carbonates) could also be due to the afore-mentioned contamination issues linked to a presence of lithogenic Cr, the latter causing

an early diagenetic overprint of the primary marine $\delta^{53}$Cr signal. Nevertheless, the observed pattern of biologically controlled Cr isotope fractionation in marine carbonates is likely a common phenomenon, as it is confirmed for all our studied sites and oceanic provinces, including also our uncontaminated samples from LEI. All these sample sets showed generally lower $\delta^{53}$Cr in marine skeletal carbonates relative to local ocean waters (Fig. 7).

**5.6. Implications for $\delta^{53}$Cr based paleo-seawater reconstructions**

Due to the complex nature of Cr isotope fractionation in the seawater-carbonate system, which is controlled by both biological/kinetic and redox processes, it is challenging to interpret the Cr isotope compositions of fossil carbonates in terms of a paleo-seawater $\delta^{53}$Cr signature. Here we use a combination of $\delta^{53}$Cr and Ce/Ce* proxies to resolve some of these issues, as the latter can be used as an independent *redox* indicator with the aim of separating (i) the redox-controlled Cr isotope effects from (ii) a local seawater Cr isotope signal. Our results from LEI suggest that the $\delta^{53}$Cr of paleo-seawater can be inferred from the intercept of the Ce/Ce* vs. $\delta^{53}$Cr correlation trend for a set of well-preserved marine carbonates (e.g., molluscan shells) collected at a specific site. A similar correlation trend between $\delta^{53}$Cr and Ce/Ce* data was also observed recently for modern ooids from Bahamas (Bonnand et al., 2013) and the intercept of this correlation also agrees with the average $\delta^{53}$Cr of Caribbean seawater (Holmden et al., 2016).

**6. Conclusions**

This study investigated the chromium isotope composition ($\delta^{53}$Cr) in present-day biogenic carbonates and seawaters collected from the southern Great Barrier Reef – Lady Elliot Island, LEI, Australia. These data from the South Pacific region are complemented by Cr isotope analysis of recent skeletal carbonates originating from the North Pacific, North and South Atlantic Oceans, Caribbean and Mediterranean Seas. Overall, the results from these globally distributed marine biogenic carbonates and local ocean waters (i.e., our and published seawater data) reveal a preferential incorporation of light Cr isotopes into marine biogenic/skeletal carbonates. However, secondary processes such as early diagenetic contamination of shells by detrital and/or lithogenic Cr phases – likely linked to Mn availability in a sediment-pore water system – can additionally shift the $\delta^{53}$Cr of marine skeletal carbonates to lower values (i.e., approaching $-0.1\permil$).

Results from LEI biogenic carbonates, with no detectable evidence for contamination, confirms the complex nature of biologically controlled incorporation of Cr into marine skeletal carbonates, which seems to be specific for different groups of calcifying organisms. In particular, molluscan shells (gastropods) showed a strong and statistically significant negative correlation between $\delta^{53}$Cr and Ce/Ce* data ($r^2 = 0.83$, $p = 0.01$). The latter can be explained by considering (i) a partial *reduction* of isotopically heavy marine Cr(VI) to Cr(III) at calcification sites, (ii) a subsequent *re-oxidation* of isotopically light Cr(III) to Cr(VI), and (iii) its final incorporation into a shell as Cr(VI)-oxyanions replacing carbonate CO$_3^{2-}$ anions. In contrast, corals show no obvious systematic relationship with respect to local seawater Ce/Ce* and $\delta^{53}$Cr values, but their Cr isotope composition tends to be generally lower and highly fractionated with respect to LEI ocean water. Interestingly, the only calcifying organism from LEI that yielded identical $\delta^{53}$Cr and Ce/Ce* to those in local seawater was a microbial carbonate (i.e., high-Mg calcite) produced by calcifying red algae (*Lithothamnion* sp.).

To conclude, different carbonate producing organisms can be used in different ways to infer the Cr isotope signature of local

seawater. Our results suggest that the latter can be constrained either from (i) an intercept of Ce/Ce* vs. $\delta^{53}$Cr correlation acquired from well-preserved molluscan shells (i.e., gastropods) and possibly ooids, or alternatively (ii) the $\delta^{53}$Cr of local seawater can also be recorded directly by microbial carbonates produced by certain species of calcifying algae (*Lithothamnion* sp.), as the latter seems to incorporate Cr isotopes without additional biological fractionation. If validated by future studies from other locations, the above multi-proxy approach could be used to reconstruct the Cr isotope signature of paleo-seawater based on $\delta^{53}$Cr and Ce/Ce* data in a set of well-preserved fossil marine carbonates (molluscan shells, microbialites, and possibly ooids) collected at a specific site.

**Acknowledgements**

This study was supported by a GACR grant (15-13310S), and partial support from the University of Adelaide Environment Institute (EI) and the ARC Linkage Project LP160101353 to JF are also greatly acknowledged. This publication is a TRaX contribution No. 404. RF acknowledges financial support via grants 11-103378 and 4181-00002B given by the Danish Agency for Science, Technology and Innovation. We thank Toni Larsen for help with ion chromatographic separations and Toby Leeper for always keeping the three TIMS at the Department of Geoscience and Natural Resource Management, University of Copenhagen, in perfect running condition. Glenn Brock (Macquarie University) is acknowledged for his help with the determination of species of marine organisms collected at Lady Elliot Island. We also thank Andreas Supper (Lady Elliot Eco Resort) and his diving team for the collection of local seawater samples; and the photographer Quinton Marais is acknowledged for the permission to use his aerial photograph of Lady Elliot Island in this article. Finally, we acknowledge the professional editorial handling by Derek Vance, and appreciate constructive comments of 4 anonymous reviewers and the editor whose feedback and advice significantly improved the quality of this study.

**Appendix A.  Supplementary material**

Supplementary material related to this article can be found online at https://doi.org/10.1016/j.epsl.2018.06.032.

**XMLVIEW: extended**

**Appendix A. Supplementary material**

The following is the Supplementary material related to this article.

begin ecomponent begin ecomponent begin ecomponent begin ecomponent begin ecomponent begin ecomponent begin ecomponent begin ecomponent begin ecomponent begin ecomponent begin ecomponent begin ecomponent begin ecomponent begin ecomponent begin ecomponent begin ecomponent begin ecomponent

Label: Supplementary tables and figures

caption: Color photographs of studied biogenic carbonates, and their taxonomic classification and mineralogy. A detailed description of analytical techniques used for mineralogical (XRD), elemental (ICP EOS, seaFAST ICP MS), and isotope analyses (TIMS) of carbonates and seawater. Complete tabulated data sets displaying elemental (Mg, Sr, Al, Mn, Fe, Cr and REE) and isotope ($\delta^{53}$Cr, $^{87}$Sr/$^{86}$Sr) variations measured in studied biogenic carbonates and seawater samples.

link: **APPLICATION : mmc1**

end ecomponent end ecomponent end ecomponent end ecomponent end ecomponent end ecomponent end ecomponent end ecomponent end ecomponent end ecomponent end ecomponent end ecomponent end ecomponent end ecomponent end ecomponent end ecomponent end ecomponent

UNCORRECTED PROOF

Please cite this article in press as: Farkaš, J., et al. Chromium isotope fractionation between modern seawater and biogenic carbonates from the Great Barrier Reef, Australia: Implications for the paleo-seawater $\delta^{53}$Cr reconstruction. Earth Planet. Sci. Lett. (2018), https://doi.org/10.1016/j.epsl.2018.06.032

**Sponsor names**

*Do not correct this page. Please mark corrections to sponsor names and grant numbers in the main text.*

**GACR**, *country*=Czechia, *grants*=15-13310S

**University of Adelaide**, *country*=Australia, *grants*=

**ARC**, *country*=Australia, *grants*=LP160101353

**Danish Agency for Science, Technology and Innovation**, *country*=Denmark, *grants*=11-103378, 4181-00002B

**Highlights**

- Cr isotopes in a seawater-carbonate system from the Great Barrier Reef, Australia.
- Systematically lower $\delta^{53}$Cr in marine biogenic carbonates relative to ocean waters.
- Coupling between $\delta^{53}$Cr and Ce/Ce* data in carbonates from Lady Elliot Island.
- Evidence for redox-controlled incorporation of Cr into marine biogenic carbonates.
- Implications for paleo-seawater $\delta^{53}$Cr reconstruction based on multi-proxy approach.

Please cite this article in press as: Farkaš, J., et al. Chromium isotope fractionation between modern seawater and biogenic carbonates from the Great Barrier Reef, Australia: Implications for the paleo-seawater $\delta^{53}$Cr reconstruction. Earth Planet. Sci. Lett. (2018), https://doi.org/10.1016/j.epsl.2018.06.032

---

## Editor Decision (ED1)

The manuscript by Frei et al. deals with application of Cr isotope ratio in deciphering paleo redox variation and modeling of Cr incorporation into the calcite lattice through an intermediate organic pathway. Authors have made reasonable effort to support their hypotheses. Both the reviewers have made reasonable suggestions for improvement of the manuscript.

I have following comments and suggestion:

A. It is not clear how much Calcium carbonate powder is required for individual Cr analyses (conc and isotope ratio measurement). Carbonate shell sampling part needs clarity. It can be written in two paragraphs. Bulk sampling and transact sampling. Is there is any information on the organic content of the shells? It would help to establish the relation between Cr concentration and TOC content since organic phases seems to be critical in Cr concentration and isotopic fractionation.

B. The manuscript sites Frakas et al (a submitted paper) frequently in various contexts. I think it should be limited to introduction only.

C. Authors have discussed Frakas et al. by siting the relation obtained between Ce anomaly and Cr isotopic fractionation. It is not required since this data is not visible to the reader and difficult to comprehend. Moreover Ce anomaly and REE content in has strong correlation with Fe-oxyhydroxide content in calcium carbonate. Many earlier papers have shown sharp change in Ce anomaly at the sed-water interface seasonally coupled with Fe mobilization. The interface redox variation may be attributed to variation in organic loading. I suggest to re- write this part without depending on Ce anomaly which is not a part of the present study.

D. The samplings of shell are carried out within 1-20 m water depth in open ocean condition. Does the seasonal water column data show any water column redox variation? No Eh or oxygen content data is available from any of the sampling site which can show possible seasonal redox variation. I other words it is not convincing that the observed 53Cr data along the shell transact is an indicator of water column redox variability.

E. The vital effect on any isotopic fractionation is very complex and less well understood compared to inorganic incorporation.

F. Finally, I would suggest reducing the length of the manuscript. It is bit laborious to search for the heart of the manuscript. Also, try to focus more on your data and model than stressing on application in paleo redox variation since it is not supported by your data.